# Semantic Uncertainty Quantification of Hallucinations in LLMs: A Quantum Tensor Network Based Method

**Pragatheeswaran Vipulanandan[1], Kamal Premaratne[1], Dilip Sarkar[2]**
[1]Department of Electrical and Computer Engineering
[2]Department of Computer Science
University of Miami
Coral Gables, FL 33146, USA
`pxv245@miami.edu, kamal@miami.edu, sarkar@miami.edu`

## Abstract

Large language models (LLMs) exhibit strong generative capabilities but remain vulnerable to confabulations, fluent yet unreliable outputs that vary arbitrarily even under identical prompts. Leveraging a quantum tensor network–based pipeline, we propose a quantum physics-inspired uncertainty quantification framework that accounts for the aleatoric uncertainty in token sequence probability for semantic equivalence-based clustering of LLM generations. In turn, this offers a principled and interpretable scheme for hallucination detection. We further introduce an entropy-maximization strategy that prioritizes high-certainty, semantically coherent outputs and highlights entropy regions where LLM decisions are likely to be unreliable, offering practical guidelines for when human oversight is warranted. We evaluate the robustness of our scheme under different generation lengths and quantization levels, dimensions overlooked in prior studies, demonstrating that our approach remains reliable even in resource-constrained deployments. A total of 116 experiments on TriviaQA, NQ, SVAMP, and SQuAD across multiple architectures (Mistral-7B, Mistral-7B-instruct, Falcon-rw-1b, LLaMA-3.2-1b, LLaMA-2-13b-chat, LLaMA-2-7b-chat, LLaMA-2-13b and LLaMA-2-7b) show consistent improvements in AUROC and AURAC over state-of-the-art baselines.

## 1 Introduction

Large language models (LLMs) have demonstrated remarkable capabilities in understanding, interpreting, and reasoning over human language and natural language generation (NLG) at a high level of fluency and coherence. LLMs, trained on massive text corpora, have led to the creation of powerful tools that brace state-of-the-art (SOTA) approaches in tasks such as summarization, question answering (Q&A), dialogue generation, and are reshaping how humans interact with machines (e.g., ChatGPT, which provides conversational AI experiences (Hosseini et al. (2023)), GitHub Copilot (Chen et al. (2021)), DALL·E (Lai et al. (2023)), and others).

As LLMs are being integrated in new applications in nearly all aspects of daily life and in almost every scientific field, the reliability of their outputs has gained increasing attention. Errors and inconsistencies they introduce can have a direct and significant impact on downstream performance. This issue is particularly evident in interactive systems such as chatbots, where the responses are expected to be not only fluent but also factually grounded (Lei et al. (2023)), especially when they are being relied upon in more sensitive application scenarios (e.g., healthcare, defense, law, journalism, autonomous driving) (Kaddour et al. (2023); Zhao et al. (2023); Lei et al. (2023)). One particular vulnerability of LLMs is *hallucinations,* which refer to LLM generations that appear coherent and plausible but are in fact incorrect, unverifiable, or entirely fabricated (Ji et al. (2023)). Hallucinated outputs can range from minor factual inaccuracies to completely spurious claims, and they pose a serious obstacle to the deployment of LLMs in high-stakes domains; use of hallucinatory content has significant ethical concerns as well.

## 1.1 PREVIOUS WORK

Most approaches for detecting and mitigating hallucinations lean heavily on supervised learning frameworks (Rateike et al. (2023); Quevedo et al. (2024)) or in-context learning techniques grounded in complex syntactic/semantic analyses (Wang et al. (2022); Manakul et al. (2023); Zhang et al. (2023)). While showing promise, their computational demands often introduce latency making them unfit for real-time applications. Numerical features (e.g., statistical patterns and structural output properties) and correlations between them may also serve as effective indicators of hallucinations (Azaria & Mitchell (2023); Lee (2023); Su et al. (2024)), yielding lightweight, resource-efficient detection methods that complement or even outperform more computationally intensive techniques. Although Bayesian deep learning (BDL) methods have long been used to quantify uncertainty in standard classification tasks (Schweighofer et al. (2024)), they are far less effective for hallucination detection in LLMs: they are challenging to calibrate, they do not reliably reflect a model's uncertainty about factual correctness, and next-token probabilities do not behave like true class-prediction probabilities. As a result, classical BDL tools struggle to capture the semantic and contextual instabilities that drive hallucinations, limiting their usefulness in this setting (Kang et al. (2025)).

Other works focus on specific types of hallucinations with particular attention paid to *confabulations,* fluent but incorrect generations that appear arbitrary (Berrios (1998)). They are sensitive to irrelevant factors (e.g., random seeds and prompt phrasing) and are distinct from other types of hallucinations, such as incorrect answers from models that have been trained on widespread misconceptions (Lin et al. (2022)), "lies" strategically generated in pursuit of reward signals in reinforcement learning setups (Evans et al. (2021)), and systematic failures triggered from flawed reasoning or generalization. To detect confabulations, Quevedo et al. (2024) employ token sequence (TS) probabilities and their induced entropy, a natural and interpretable measure of uncertainty (Lindley (1956); Kadavath et al. (2022)), but naive estimates of entropy may not serve as a good indicator of hallucinations (Xiao et al. (2020)) because lexical/syntactic differences in LLM generations often do not imply a semantic difference. Therefore, Farquhar et al. (2024) and Kossen et al. (2024) employ *semantic entropy (SE),* not lexical/syntactic variations, of generated outputs to quantify hallucinatory behavior. The output (TSs) generated when an LLM confronts the same input (context + prompt) repeatedly are clustered based on their bidirectional entailment (as assessed by DeBERTa). Higher semantic entropy indicates higher likelihood of hallucination. In a similar vein, Kernelized Likelihood Entropy (KLE) (Nikitin et al. (2024)) employs a graded similarity measure to account for semantic differences; Semantic Nearest Neighbour Entropy (SNNE) (Nguyen et al. (2025)) aggregates pairwise similarities with LogSumExp smoothing, avoiding explicit clustering and improving robustness on summarization and translation; Semantic Density (SD) (Qiu & Miikkulainen (2024)) weights semantic distances by generation probabilities. These approaches improve confabulation detection, but are sensitive to clustering quality, similarity functions, and TS probability fluctuations.

Structure-aware and perturbation-based methods have also been explored for detecting hallucinations. Graph Uncertainty (Jiang et al. (2024)) and KEA (Kernel-Enriched AI) explain (Haskins & Adams (2025)) model relationships between knowledge graphs (KGs) constructed from LLM outputs and ground truth claims for detection of hallucinations. But they incur heavy computational overhead and rely on an external knowledge base and accurate KG construction. Sampling with Perturbation for Uncertainty Quantification (SPUQ) (Gao et al. (2024)) and Semantic-Invariant Perturbation Sampling (Cox et al. (2025)) estimate uncertainty by probing model sensitivity to paraphrased or perturbed inputs, exposing prompt sensitivity but at the cost of repeated queries.

In spite of the advances they have made, SOTA methods still struggle with reliably detecting fabricated content across diverse tasks and domains (Li et al. (2023)). A possible reason is that they leave unaddressed the aleatoric uncertainty of the TS probabilities. Hallucination risk should depend on how sensitive these TS probabilities, and hence the semantic entropies they induce, are to model perturbations. This highlights the need for *local,* rather than *global,* sensitivity metrics to assess TS probability uncertainty. Prior studies have also not examined the robustness of their approaches with different quantization levels despite the latter playing a major role in real-world deployment of LLMs.

## 1.2 OUR CONTRIBUTIONS

"White box" inherently interpretable approaches (e.g., random sampling-based variational inference methods (Ghahramani (2015))) can be computationally prohibitive for UQ of massively scaled LLMs

and physics-inspired methods have emerged as a compelling alternative. Our work employs the physics-inspired approach, advocated in Principe (2010), and adopted in Singh & Principe (2020) and Vipulananthan et al. (2024), that leverages perturbation theory of an appropriate quantum wave function to offer a deterministic, one-shot, interpretable (Rudin (2019); Linardatos et al. (2021)) method for quantifying uncertainty locally and at different resolutions.

The main contributions and key advantages of our approach are as follows: **(a)** We introduce *TS probability uncertainty* as a novel indicator of confabulation detection. For this purpose, we view the TS probabilities as the wave function of a quantum tensor network (QTN) (Qi & Ranard (2019); Vipulananthan et al. (2024); Vipulanandan et al. (2026)) and leverage perturbation theory for UQ. **(b)** We offer a method, based on entropy maximization, to calibrate the TS probabilities to account for this uncertainty associated with TS probabilities. **(c)** This leads us to cluster LLM generations based on not only SE but uncertainty of TS probabilities. So, instead of simply flagging high entropy generations as hallucinations (Farquhar et al. (2024)), we offer a more meaningful scheme that flags outputs associated with higher uncertainty so that one may reduce the likelihood of false negatives and false positives. **(d)** We evaluate the robustness of our framework under different LLM quantization levels as well as under different generation lengths, ranging from short phrase answers to sentence-level outputs. Our results show that the method is robust and remains reliable even in resource-constrained or efficiency-optimized deployments, thus extending the utility of hallucination detection to real-world settings. The overall flow of our approach is illustrated in Fig. 1.

## 2 PROPOSED METHOD FOR UQ OF LLM GENERATIONS

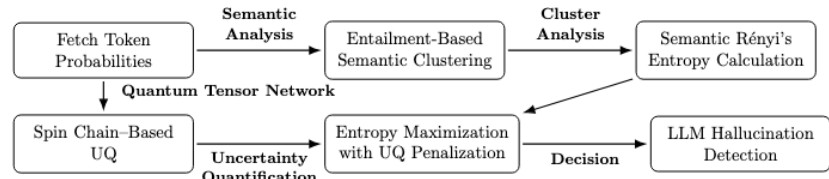

Figure 1: Overview of our hallucination detection pipeline. Sequences are clustered via directional entailment, and UQ obtained through QTN is used for entropy maximization to enable reliable LLM hallucination detection.

### 2.1 PRELIMINARIES: SEMANTIC CLUSTERING

Consider the output TS $\underline{s}$ that the LLM generates in response to input $\underline{y}$. Let $s_i$ denote the $i$-th token and $\underline{s}_{<i}$ the sequence of previous tokens. Then, the probability of $\underline{s}$, conditioned on $\underline{y}$, is

$$P(\underline{s} \mid \underline{y}) = \prod_i P(s_i \mid \underline{s}_{<i}, \underline{y}). \tag{1}$$

Clustering is determined by the semantic dissonance between output TSs as determined by bidirectional entailment (for which we employ DeBERTa). Let $C$ denote the set of semantically dissonant classes and $|C|$ its cardinality. Treating cluster label $c_j$ as a random variable (on the space of $C$), the *cluster probability* that the TS $\underline{s}$ belongs in $c_j \in C$ and the induced *semantic (Shannon) entropy* are

$$p_c^{(j)} \triangleq P(c_j \mid \underline{y}) = \frac{\sum_{\underline{s} \in c_j} P(\underline{s} \mid \underline{y})}{\sum_{j=1}^{|C|} \sum_{\underline{s} \in c_j} P(\underline{s} \mid \underline{y})}; \ \ \mathrm{SE}_S(\underline{y}) = -\sum_{c_j \in C} p_c^{(j)} \log p_c^{(j)}, \tag{2}$$

respectively. When the TS probabilities $P(s_i \mid \underline{s}_{<i}, \underline{y})$ are available, $p_c^{(j)}$ are computed from equations 1 and 2; otherwise, we use $p_c^{(j)} \approx \mathbb{I}(c = c_j)/|C|$, the membership count in each cluster. The latter is referred to as *discrete semantic (Shannon) entropy* in Farquhar et al. (2024).

### 2.2 SEMANTIC RÉNYI ENTROPY

The physics-inspired framework for UQ in Principe (2010) requires a non-parametric feature mapping of the TS probabilities in the form of its kernel mean embedding (KME) in the reproducing kernel

Hilbert space (RKHS) determined by a suitable positive semi-definite kernel (Aronszajn (1950); Schölkopf et al. (2015)). We employ the Gaussian kernel $\kappa_\sigma(p_s; x)$, where $p_s \triangleq P(\underline{s} \mid \underline{y})$ and $x \in \mathbb{R}$. The resulting KME and its empirical estimate are, respectively,

$$\psi_{\underline{y}}(x) = \int_{\underline{s}} \kappa_\sigma(p_s, x)\, p_s\, dp_s; \quad \widehat{\psi}_{\underline{y}}(x) = \frac{1}{R} \sum_{r=0}^{R-1} \kappa_\sigma(p_s^{(r)}; x)\, p_s^{(r)}, \tag{3}$$

where $\{\underline{s}^{(r)}, r = 0, \ldots, R-1\}$ denotes the LLM generations when the input $\underline{y}$ is repeated $R$ times and $p_s^{(r)} \triangleq P(\underline{s}^{(r)} \mid \underline{y})$. As Principe (2010) establishes, $\widehat{\psi}_{\underline{y}}(\textbf{.})$ serves as a kernel-based empirical estimator of the more general notion of Rényi (quadratic) entropy

$$\mathrm{SE}_R(\underline{y}) = -\log \sum_{j=1}^{|C|} p_c^{(j)^2} = -\log\left(\mathbb{E}[p_c]\right), \tag{4}$$

which in turn is a direct measure of uncertainty in a distribution. In this work, we exclusively focus on the quadratic form of Rényi entropy and, for brevity, refer to it throughout the paper as semantic R'enyi entropy. Here, $\mathbb{E}[\textbf{.}]$ denotes expectation (over the space of $C$). Indeed, $\widehat{\psi}_{\underline{y}}(x)$ can be viewed as how uncertainty varies with $x$. For purposes of UQ, we view this KME — to be precise, a sampled version $\{\widehat{\underline{\psi}}_{\underline{y}}\}$ of $\widehat{\psi}_{\underline{y}}(x)$ — as a 'data' wave function associated with the time-independent Schrödinger equation of a QTN (Qi & Ranard (2019); Vipulananthan et al. (2024); Vipulanandan et al. (2026)).

Given that it is Rényi entropy which underpins our physics-inspired framework for UQ of TS probabilities, we opt to use the same (instead of Shannon entropy as in SOTA methods) for assessing semantic entropy of output clusters associated with LLM generations too.

### 2.3 Accounting for Uncertainty in TS Probabilities

**UQ of TS Probabilities** For purposes of UQ of TS probabiltiies, we identify a QTN whose Hamiltonian $\widehat{\underline{H}}$ has $\{\widehat{\underline{\psi}}_{\underline{y}}\}$ as one of its eigen-modes, and apply perturbation theory to compute the corrections (we only look at the first-order corrections) to all the eigen-modes/energies of $\widehat{\underline{H}}$. Then, we construct the first-order uncertainty 'feature' vectors

$$\underline{V}_m^{(1)}(x) = E_m^{(1)} + \frac{\sigma^2}{2} \frac{\nabla_m^2 |\psi_m^{(1)}(x)|}{|\underline{\psi}_m^{(1)}(x)|}, \text{ where } E_m^{(1)} = -\min_{p_x} \frac{\sigma^2}{2} \frac{\nabla_m^2 |\underline{\psi}_m^{(1)}(x)|}{|\underline{\psi}_m^{(1)}(x)|}. \tag{5}$$

Here, $\{\underline{\psi}_m, E_m\}$ and $\{\underline{\psi}_m^{(1)}, E_m^{(1)}\}$, for $m \geq 0$, are the eigen-pairs of the Hamiltonian $\widehat{\underline{H}}$ ordered as $E_0 \leq E_1 \leq \cdots$, and their first-order correction terms, respectively. For each mode $m \geq 0$, the Laplacian $\nabla_m^2 |\underline{\psi}_m^{(1)}(x)|$ measures the change in the first-order correction from its average in the local neighborhood across the modes and the vector $\underline{V}_m^{(1)}(x)$ can be viewed as a 'spectrogram' of uncertainties across TS probability amplitudes. If $p_s^{(r)}$ is mapped to $x^{(r)}$ in the RKHS, the uncertainty $\mathrm{UQ}(p_s^{(r)})$ associated with the TS probability $p_s^{(r)}$ is taken as

$$\mathrm{UQ}(p_s^{(r)}) = \frac{1}{M} \sum_m \underline{V}_m^{(1)}(x)\Big|_{x=x^{(r)}}, \tag{6}$$

where the summation is carried over $M$ (we use $M = 8$) modes that are adjacent to $\{\widehat{\underline{\psi}}_{\underline{y}}\}$. Additional details on QTN based UQ are provided in Appendix A.2.

**Calibrated Adjustment of TS Probabilities** We now adjust the TS probabilities as

$$p_s^{(r)*} = \arg\max_{\widehat{p}_s^{(r)}} \left\{ -\log\left(\widehat{p}_s^{(r)^2} + (1 - \widehat{p}_s^{(r)})^2\right) - \lambda \cdot \frac{1}{\mathrm{UQ}(p_s^{(r)})} \cdot \mathrm{KL}\left(\widehat{p}_s^{(r)} \| p_s^{(r)}\right) \right\}. \tag{7}$$

The $\log(\textbf{.})$ term in the right-hand side is the Rényi entropy; the second term penalizes the Kullback–Leibler (KL) divergence between the adjusted probability $\widehat{p}_s^{(r)}$ and $p_s^{(r)}$ scaled inversely with

the uncertainty $\mathrm{UQ}\left(p_s^{(r)}\right)$; the hyperparameter $\lambda > 0$ controls the trade-off between entropy maximization and the adjusting the TS probability. In essence, we are maximizing entropy of the TS probabilities while penalizing the error between $p_s^{(r)}$ and its adjusted value $\widehat{p}_s^{(r)}$. This scheme which appears in Fig. 1 enables one to account for uncertainty associated with TS probabilities so that a more informed decision regarding existence/absence of hallucinatory behavior could be made. Appendix D contains additional discussion of computational overheads together with the pseudocode implementations of key components.

## 2.4 Intuition Behind the Proposed Approach

Our method integrates perturbation-based uncertainty quantification and maximum entropy inference into a unified framework for correcting TS probabilities in a principled manner.

**Perturbation-based uncertainty quantification.** To estimate uncertainty in the TS probabilities, we map the KME of the TS probability distribution into the eigen structure of a QTN. The empirical KME is embedded as an eigen-mode of an admissible Hamiltonian $\widehat{H}$ associated with the QTN. Therefore, perturbing $\widehat{H}$ corresponds to perturbing the underlying TS probabilities; the resulting first-order corrections to its eigen-modes/energies quantify how sensitive the modeled distribution is to infinitesimal changes in its inputs. This mirrors standard quantum mechanical perturbation theory where the instability of an eigen-state under perturbation reveals its local variability.

Large first-order corrections, therefore, indicate highly unstable TS probabilities; small corrections indicate locally stable regions. This produces an interpretable, physically grounded uncertainty measure, formalized in eq. (6), that captures the *local variability* of the TS probabilities in the RKHS amplitude domain.

**Maximum entropy inference under partial knowledge.** When only partial information is available, the maximum entropy principle provides the least biased estimate of an unknown probability distribution. Among all distributions consistent with the known constraints, the one with the highest entropy is maximally non-committal with respect to the missing information (Jaynes (1957; 2003)). Equivalently, the maximum entropy distribution minimizes its KL divergence from the uniform distribution. This idea has deep roots in robust statistics and variational inference, where uncertainty in the likelihood leads naturally to regularization toward higher-entropy, less overconfident solutions.

Our formulation implements precisely this. The Rényi entropy term in eq. (7) specifies how much the probabilities *should* be lifted to remain maximally non-committal, whereas the KL term ensures that the adjusted probabilities stay close to the model's empirical TS estimates. When uncertainty is high, the KL penalty relaxes and the entropy term dominates; when uncertainty is low, the KL term keeps the adjusted probabilities anchored near their empirical values.

**Bringing the two components together.** The overall procedure thus follows a coherent logic: (i) infer local uncertainty in the TS probabilities by perturbing the Hamiltonian and reading out the first-order eigen-mode corrections; (ii) use these corrections to scale the KL penalty in the entropy maximization step; and (iii) adjust the TS probabilities to the maximum entropy distribution allowed by the empirical TS estimates. High-uncertainty regions are pushed toward higher entropy, while low-uncertainty regions remain close to the model's original predictions.

This yields a deterministic, single-shot, principled method for correcting TS probabilities in accordance with their underlying uncertainty, unifying perturbation-based UQ with maximum-entropy reasoning in a single framework.

## 3 Experiments

**Datasets** Our evaluations cover Q&A in trivia knowledge (TriviaQA (Joshi et al. (2017))), general knowledge (SQuAD 1.1 (Stanford NLP Group)), and open-domain natural questions (NQ-Open (Lee et al. (2019))) derived from actual queries to Google Search (Kwiatkowski et al. (2019)), and mathematical word problems (SVAMP (Patel et al. (2021)). Only the results summary is reported here; while detailed results are deferred to Appendix C.

**Models** We use a diverse set of LLMs obtained via Huggingface (`huggingface.com`), including Falcon-RW 1B (Penedo et al. (2023)), LLaMA-3.2 1B (Grattafiori et al. (2024)), LLaMA-2-7B-chat, LLaMA-2 7B, LLaMA-2-13B-chat, LLaMA-2 13B (Touvron et al. (2023)), Mistral-7B-instruct-v0.3, and Mistral-v0.1 7B (Jiang (2024)). The inclusion of smaller-scale models and quantized versions of larger models serves two purposes. From a practical standpoint, resource limitations necessitate the use of lighter models to enable extensive sampling and repeated querying. From a scientific viewpoint, evaluating smaller and compressed models allows us to explore hallucinations and uncertainty under constrained model capacity and precision, providing insights into how semantic entropy behaves not only in ideal settings but also in more realistic, resource-constrained deployments. This is particularly relevant as smaller and quantized models are increasingly being deployed in edge, mobile, and enterprise environments. All experiments [1] were run on a workstation with 64 GB memory and NVIDIA A6000 GPU.

**Model Prompting, Entailment, and Answer Selection** Appendix B provides the prompt template used for all datasets to generate LLM responses and the prompt we used to detect entailment. While other classifiers could have been used, we employ a DeBERTa-large model fine-tuned on the MNLI dataset (Williams et al. (2017)) for entailment prediction. This method builds upon previous research in paraphrase detection based on embedding similarity (Socher et al. (2011); Yu et al. (2014)) and BERT-style models (He et al. (2020); Tay et al. (2021)). For simplicity, entailment is checked by concatenating the question with one answer and comparing it to the concatenation of the question with another answer. Instruction-tuned LLMs, such as LLaMA 2, GPT-3.5 Turbo or GPT-4, could also have been used to predict entailment between generated outputs.

**Entropy Maximization** Following UQ of the LLM generations, entropy maximization (see equation 7) was applied to identify the 'optimum' answer based on the associated uncertainty estimates. The effectiveness of this approach (in the sense of whether incorporating uncertainty information improves hallucination detection) was assessed using the evaluation metrics described below. We also investigated the optimal choice of the hyperparameter $\lambda$. Implementation details and extended analyses are deferred to Appendix C.3.

**Comparison Methods** For evaluation, we use the following methods for comparison against the proposed Rényi semantic entropy pre- and post-TS uncertainty integration ($SE_R$ and $SE_R^+$, respectively): naive entropy (NE), semantic entropy ($SE_S$), and discrete semantic entropy ($DSE_S$), all employed in Farquhar et al. (2024), and two strong baseline methods, embedding regression (ER) and verifier $p(True)$. ER is a supervised method inspired by the P(IK) approach (Kadavath et al. (2022)). Rather than fine-tuning the entire LLM as in the original work, this method trains a logistic regression classifier on the final hidden representations to predict answer correctness, improving both simplicity and reproducibility. This baseline performs well with in-distribution data but poorly with out-of-distribution data. In $p(True)$ (Kadavath et al. (2022)), the model samples multiple candidate answers and is then asked, via a multiple-choice prompt, whether the top answer is "True" or "False". Confidence is measured by the probability assigned to the "True" response. A few-shot strategy, using up to 20 labeled examples, further enhances performance, although context window limitations sometimes require reducing the number of few-shot examples.

**Evaluation Metrics** We use three primary metrics — each based on an automated assessment of factual consistency with reference answers from the datasets employed — for evaluation purposes: **(a)** *AUROC (Area Under the Receiver Operating Characteristics)* measures the ability of a classifier to distinguish between correct and incorrect answers while accounting for both precision and recall. Intuitively, it represents the probability that a randomly selected correct answer receives a higher confidence score than a randomly selected incorrect one. A perfect model achieves an AUROC of 1. **(b)** *RAC (Rejection Accuracy Curve)* measures the question-answering accuracy that a specified percent of the most confident model predictions. An effective uncertainty estimation approach should ensure that the high-confidence predictions being retained are more accurate, with RAC improving as less confident examples are progressively excluded. **(c)** *AURAC Area Under the Rejection Accuracy Curve)* quantifies how accuracy changes as increasingly uncertain predictions are rejected. Higher AURAC values indicate that the uncertainty method more effectively distinguishes accurate from

---

[1] code available: https://github.com/pragasv/semantic-entropy-UQ-.git

inaccurate responses. Unlike AUROC, AURAC is directly sensitive to the model's overall accuracy, providing a complementary perspective on the quality of uncertainty estimation methodology.

# 4 RESULTS

## 4.1 UNCERTAINTY-AWARE SEMANTIC ENTROPY: A WORKED EXAMPLE

This section presents a worked example illustrating the computation of Semantic Rényi entropy with UQ maximization (SRE-UQ), denoted as $\mathrm{SE}_R^+$, together with NE and SE. The query used for this illustration is, *"Which oil producer is a close ally of the United States?"* repeated ten times. The corresponding TS probabilities are in Table 1.

The results demonstrate that incorporating uncertainty quantification into semantic entropy (i.e., $\mathrm{SE}_R^+$) yields systematically lower entropy estimates relative to both NE and 'standard' Shannon entropy-based $\mathrm{SE}_S$. This reduction arises because semantically equivalent generations are clustered together for this question, while their TS probabilities exhibit reduced aleatoric variability. Importantly, accounting for such uncertainty mitigates overestimation biases in entropy, providing a more faithful measure of confabulation in LLM outputs.

| LLM Generation | Cluster ID | $\widetilde{p}_s^{(r)}$ | $p_s^{(r)}$ | $p_c^{(j)}$ | $p_c^{(j)*}$ | NE | $\mathrm{SE}_S$ | $\mathrm{SE}_R^+$ |
|---|---|---|---|---|---|---|---|---|
| Russia | 1 | 0.05899 | 0.01814 | 0.01814 | 0.02223 | -0.03159 | -0.03159 | 0.00049 |
| Saudi Arabia | 2 | 0.57761 | 0.17765 | 0.88824 | 0.85880 | -0.13331 | -0.04572 | 0.73754 |
| Saudi Arabia | 2 | 0.57761 | 0.17765 | 0 | 0 | -0.13331 | – | – |
| Iran | 3 | 0.07227 | 0.02223 | 0.02223 | 0.02697 | -0.03674 | -0.03674 | 0.00073 |
| Saudi Arabia | 2 | 0.57761 | 0.17765 | – | – | -0.13331 | – | – |
| Kuwait | 4 | 0.08940 | 0.02749 | 0.02749 | 0.03488 | -0.04291 | -0.04291 | 0.00122 |
| Qatar | 5 | 0.02185 | 0.00672 | 0.00672 | 0.01214 | -0.01460 | -0.01460 | 0.00015 |
| Saudi Arabia | 2 | 0.57761 | 0.17765 | – | – | -0.13331 | – | – |
| Iraq | 6 | 0.12086 | 0.03717 | 0.03717 | 0.04498 | -0.05315 | -0.05315 | 0.00202 |
| Saudi Arabia | 2 | 0.57761 | 0.17765 | 0 | 0 | -0.13331 | – | – |
| **Total** | | 3.25144 | 1.00000 | 1.00000 | 1.00000 | **0.84557** | **0.22471** | **0.12951** |

Table 1: An example calculation of TS probability and uncertainty metrics corresponding to the query above. Columns: $\widetilde{p}_s^{(r)}$ are the raw (unnormalized) sequence probabilities; $p_s^{(r)}$ are the normalized sequence probabilities (equation 3); $p_c^{(j)} = P(c_j \mid \underline{y})$ are the cluster probabilities (equation 2) post-normalized; $p_c^{(j)*}$ are the cluster probabilities post-TS uncertainty integration (7). Methods: NE and $\mathrm{SE}_S$ denote naive entropy and semantic entropy, respectively (Farquhar et al. (2024)); $\mathrm{SE}_R^+$ denotes semantic Rényi entropy post-TS uncertainty integration. Values for only one representative LLM generation per cluster are shown.

Table 1 illustrates this effect for the example query. It compares the semantic response distributions produced by 'standard' Rényi entropy-based $\mathrm{SE}_R$ and its uncertainty-adjusted variant $\mathrm{SE}_R^+$. While several hallucinated outputs (e.g., Qatar, Iraq, Iran) receive non-trivial probability mass, $\mathrm{SE}_S$ underestimates the associated confabulation risk. In contrast, the proposed uncertainty-aware adjustment penalizes responses with higher uncertainty, redistributing probability mass toward more semantically coherent answers. For instance, Saudi Arabia—a contextually appropriate response—receives a higher probability assignment after adjustment. This highlights a promising direction: even when hallucinations are present, localized uncertainty quantification enables prioritization of high-certainty, semantically meaningful answers, a factor not considered in prior work.

## 4.2 DETECTING CONFABULATIONS

**Benchmark Datasets.** Building on the worked example above, we now present the results obtained from TriviaQA, NQ, SVAMP, and SQuAD datasets, across multiple LLM models such as Mistral 7B, LLaMA-2 7B, LLaMA-3.2 1B, Falcon-rw-1B and LLaMA-2 13B as well as uncertainty estimation methods; detailed validation results appear in Appendix C.

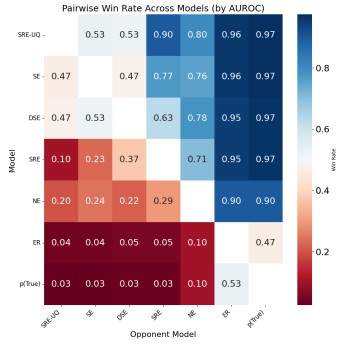
(a) AUROC-based win rate.

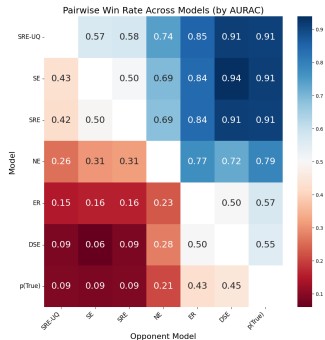
(b) AURAC-based win rate.

Figure 2: Pairwise win rate matrices across SOTA hallucination detection methods on diverse datasets and LLM models. This summarizes 116 experimental scenarios. Each cell indicates the probability that the row model outperforms the column model. Semantic Rényi entropy with UQ maximization consistently outperforms baselines, even surpassing methods reliant on supervised learning.

Fig. 2(a) reports the SRE-UQ's performance against SOTA methods evaluated via AUROC performance. Higher AUROC scores indicate better separability between factually correct and hallucinated outputs. Across all evaluated models, SRE-UQ is competitive with SOTA baseline methods. Notably, SRE-UQ achieves better AUROC despite not relying on ground truth labels for training. This is evidence of its effectiveness in hallucination detection with no reliance on supervised fine-tuning.

While AUROC reflects a model's global ranking ability, practical deployments often involve thresholding on confidence scores to reject outputs associated with higher uncertainty. To assess this, we compute the RAC over rejection thresholds ranging from 80% to 100% confidence. Fig. 2(b) presents the SRE-UQ's performance against SOTA methods evaluated via AURAC performance. Consistently, SRE-UQ maintains strong AURAC as more uncertain outputs are filtered out. Particularly it offers superior robustness compared to discrete semantic entropy, naive entropy, and supervised p(True) baseline. This is evidence of its effectiveness in pruning predictions that are associated with higher uncertainty. Appendix C provides in-depth results at the dataset level for Trivia-QA, SQuAD, NQ-Open, and SVAMP.

**Quantized LLMs**   While most prior work on hallucination detection benchmarks full-precision models, real-world deployments almost always rely on quantization (e.g., 16-bit, 8-bit, or 4-bit) to reduce inference latency and memory consumption. However, quantization does not merely compress model weights — it also perturbs probability distributions and modifies token-level uncertainty. These shifts can affect semantic entropy measurements and alter the relative competitiveness of different detection methods. Thus, a rigorous study of hallucination detection must examine the stability of detection methods across quantization levels, ensuring that performance gains are not an artifact of precision settings but persist in practical deployment regimes. Figure 3 presents pairwise win rate matrices of hallucination detection methods under different quantization settings (16-bit, 8-bit, and 4-bit), evaluated using both AUROC and AURAC metrics. The results show that SRE-UQ maximization remains robust across quantization levels, consistently outperforming or matching SOTA baselines even under aggressive compression.

The quantization-wise analysis reveals that SRE -UQ maximization consistently outperforms alternative detection strategies across 16-bit, 8-bit, and 4-bit models. While quantization slightly reduces absolute AUROC values due to coarser uncertainty calibration, the relative ordering of methods remains largely stable, demonstrating that the proposed approach is resilient to reduced precision. This robustness is critical for deployment at scale, where low-bit quantization is increasingly adopted to balance efficiency with reliability in safety-sensitive applications.

## 4.3   Entropy Uncertainty Analysis

To further assess the effect of uncertainty-aware adjustments proposed in equation 7, we analyze how SRE changes across different levels of generation diversity. Here, we measure the number of distinct semantic clusters produced by the model when answering the same question multiple times

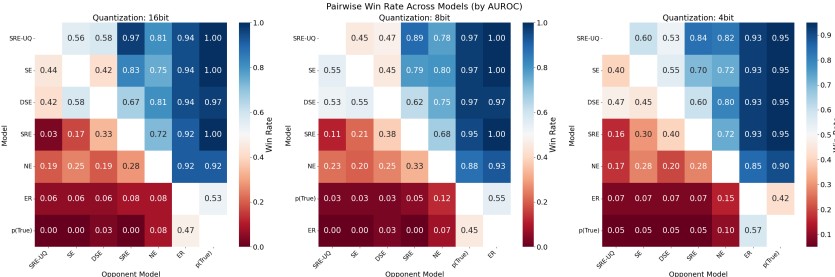

(a) Pairwise AUROC-based win rate matrices across quantization levels.

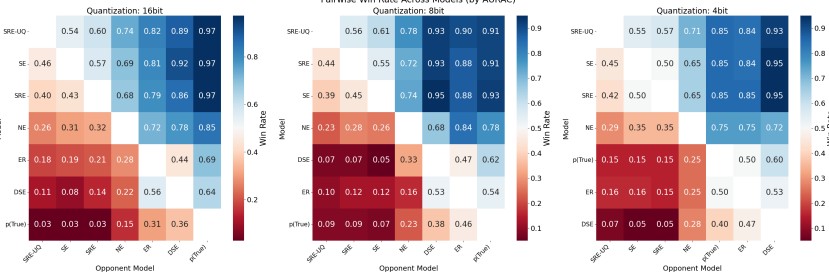

(b) Pairwise AURAC-based win rate matrices across quantization levels.

Figure 3: Evaluation of SOTA hallucination detection methods across different quantization levels. This summarizes 116 experimental scenarios. Each subfigure shows pairwise win rate matrices under a different quantization precision (16-bit, 8-bit, 4-bit). SRE-UQ maximization consistently outperforms baselines across both AUROC and AURAC, demonstrating robustness in hallucination detection against reduced precision.

(10 times). As shown in Fig. 4, the x-axis represents the observed entropy for a given question. A value of 0 on the x-axis indicates that all sampled generations were semantically identical - hence entropy is zero. This analysis directly probes how model uncertainty and semantic variability interact under the proposed adjustment mechanism.

While much of the prior work in hallucination detection relies on applying a global threshold to entropy-based scores, such approaches ignore how model behavior varies across different ranges of historical entropy. To address this, we evaluate how the change in entropy (normalized difference) distributes across old entropy bins for each LLM. This perspective provides a more fine-grained decision-making criterion: instead of arbitrarily choosing a fixed threshold, one can calibrate decisions depending on the region of the entropy spectrum.

The results demonstrate that entropy change is not uniform across the input space: low-entropy regimes (close to zero) remain relatively stable, while intermediate ranges (particularly 0.25–0.75) are highly volatile. This implies that decision rules based solely on absolute entropy thresholds are brittle and may fail to capture these nuanced behaviors. Instead, practitioners should treat certain entropy regions with caution and, in some cases, defer to auxiliary signals before committing to a prediction.

## 5 CONCLUSION

This work presents a novel framework for hallucination detection in LLMs which leverages semantic Rényi entropy and a quantum TN-based UQ method that explicitly models and accounts for aleatoric uncertainty of sequence probabilities. Through extensive evaluations on several datasets, diverse LLM architectures, and various quantization precision (16-bit, 8-bit, and 4-bit), we demonstrate that semantic entropy measures, especially when combined with principled uncertainty penalization, significantly improve the detection of hallucinated outputs. By adjusting output probabilities to maximize entropy while penalizing deviations weighted by uncertainty, the proposed method yields consistent gains in AUROC, AURAC and RAC across 116 experiments, outperforming strong

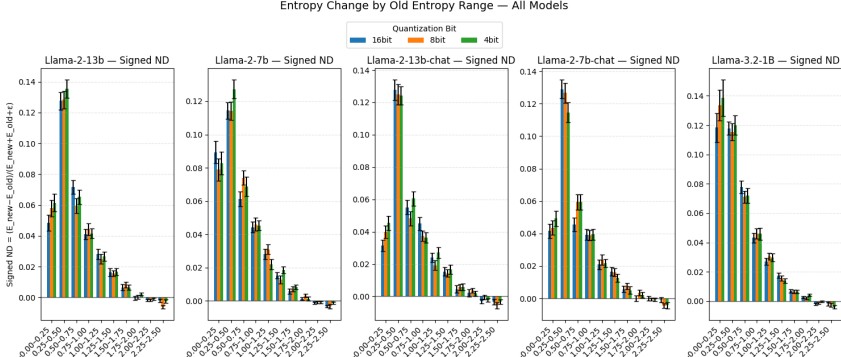

Figure 4: Signed normalized difference (ND) in entropy across old entropy bins for LLaMA-2 variants under different quantization settings (16-bit, 8-bit, 4-bit). Bars represent the mean signed ND with error bars indicating standard error. Notably, all models exhibit the largest variability in the 0.25–0.50 entropy range, suggesting this is a high-risk region where models frequently oscillate between multiple confident yet semantically divergent answers.

baselines, without requiring supervised fine-tuning. Additionally, this provides a principled approach to select answers from an LLM even when it is hallucinating.

Incorporating uncertainty not only sharpens the sensitivity of entropy-based metrics but also enables principled answer selection, even under confabulation. By adjusting TS probabilities to balance entropy maximization with uncertainty-aware penalization, our method promotes more coherent and trustworthy outputs. Furthermore, the uncertainty quantification method operates utilizing QTNs and perturbation theory, making it suitable for scalable, explainable real-world deployments.

Overall, this study highlights the importance of uncertainty-aware semantic evaluation in achieving reliable language generation, and opens up new avenues for integrating information-theoretic principles with uncertainty quantification to advance the robustness of LLMs.

**Limitations** While the proposed method combining quadratic semantic Rényi entropy and quantum TN-based UQ shows great promise in detecting hallucinations, a few limitations remain. Most notably, our evaluations are conducted using relatively small language models due to computational resource constraints. While these models enable efficient experimentation, they may not fully capture the behavior of larger frontier models such as GPT-4 or Claude. Consequently, both the absolute detection accuracy and the complexity of hallucinations observed could differ at larger scales. Furthermore, our semantic clustering relies on an external entailment predictor (DeBERTa-large fine-tuned on MNLI), and any inaccuracies in this entailment model may propagate into the entropy estimates. Finally, the proposed approach requires access to token-level probabilities, which restricts its applicability to open-weight models; black-box LLMs, where such information is not exposed, remain outside the scope of this method.

The uncertainty of TS probabilities in equation 6 may very well serve to arrive at a meaningful definition of an uncertainty associated with each semantic cluster. For instance, for the cluster $C_j$, one could employ $\mathrm{UQ}\left(C_j \mid \underline{y}\right) = (1/|C_j|) \sum_{s^{(r)} \in C_j} \mathrm{UQ}\left(p_s^{(r)}\right)$. This may facilitate one to identify semantic clusters whose membership is more sensitive to changes in TS probabilities which in turn may lead to practical guidelines for identifying situations where automated outputs should be supplemented with careful, perhaps human-assisted, review. We hope to undertake this direction of research in the future.

ACKNOWLEDGMENTS

This work is based on research supported by National Science Foundation (NSF) award #2530256.

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

## A  Expanded Version of Section 2

### A.1  Semantic Equivalence-Based Clustering

Computation of semantic entropy requires clustering LLM-generated output TSs, such that the TSs in each cluster are *semantically equivalent*, while those in different clusters are *semantically dissonant*. To identify such semantically equivalent TSs, Farquhar et al. (2024) employ *bidirectional entailment*, a notion that has attracted considerable attention in natural language inference (NLI). We, use pretrained models such as DeBERTa-Large-MNLI or general-purpose language models like GPT-3.5 to assess entailment between sentence pairs. When confronted with a new TS, we check for bidirectional entailment with a representative TS from each cluster. If no entailment is found with any of these, a new cluster is created. This process of semantic clustering allows a semantic entropy to be computed for the clusters associated with each question. Higher entropy indicates higher likelihood that the LLM is hallucinating when responding to that *question*. The top rung of Fig. 1 summarizes this process in Farquhar et al. (2024).

However, semantic entropy itself is not a reliable measure for identifying hallucinations because TS probabilities can be highly sensitive to model perturbations. This warrants adjusting the confidence with which hallucinatory behavior is deemed to be present or absent. We therefore need *local* information (as opposed to *global* metrics) that allows us to assess sensitivity of the TS probability of each LLM generation.

### A.2  Quantum TN-Based UQ Method for Deep Learning Models

Physics-inspired methods offer such local uncertainty information and have emerged as a compelling way for UQ of deep learning models (Singh & Principe (2020; 2021); Singh et al. (2024)). However, these methods are tethered to the Hamiltonian of the quantum harmonic oscillator (QHO) and are hence severely constrained in their application. In contrast, the recent work in Vipulananthan et al. (2024) introduces a deterministic, interpretable, and single-shot UQ method built on a QTN-based Hamiltonian. Moreover, it leverages the well-established method of perturbation theory to ferret out multi-resolution, local uncertainty estimates that are ideally suited for our current purpose.

In our work, we expand this QTN-based framework—originally developed for time-series signals—to the domain of semantic datasets, where TS probability distributions can be naturally viewed as signal-like processes. This extension allows us to capture semantic uncertainty in natural language through the same principled, physics-inspired lens. The importance of this contribution is twofold: first, it bridges the gap between temporal signal analysis and semantic modeling, demonstrating that techniques rooted in quantum physics can generalize beyond time-series to complex language domains. Second, by bringing interpretability and multi-resolution uncertainty estimates into the study of semantic entropy, our approach lays the groundwork for more trustworthy and explainable AI systems, especially in high-stakes applications where hallucination detection and careful decision-making are critical. Fig. 5 shows the QTN-based UQ method that we leverage in our work.

### A.3  Rationale for Adopting Semantic Rényi Entropy

While SE is widely used for characterizing uncertainty in LLM outputs, its inherent properties introduce limitations in the present setting. For any fixed predictive distribution, SE is always greater than or equal to quadratic SRE (Harremöes (2009)), but this pointwise ordering does not in general determine the relative spread of the two measures across a corpus of examples. Empirically, however, SE exhibits a larger variation in most cases (between hallucinatory and non-hallucinatory behavior) in our experiments, which correlates with slightly better AUROC performance of SE (see Fig. 2(a)) (Hand & Till (2001)). SE also emphasizes rarer outcomes. But the tail probabilities do not play a role in AURAC (see Fig. 2(b)) which measures how well increasing entropy corresponds to decreasing correctness.

A more decisive consideration arises from the structural compatibility between SRE and the proposed UQ framework. Quadratic Rényi entropy is a second-order functional of the underlying probability distribution and is directly estimated by the KME. This correspondence enables the KME to be encoded as an eigen-mode of a QTN Hamiltonian. Perturbations of this Hamiltonian, in turn, yield first-order and higher-order spectral corrections that quantify local sensitivity in the amplitude domain.

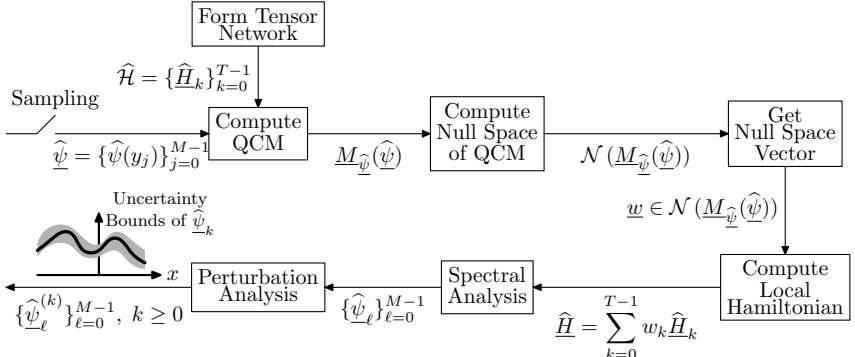

Figure 5: Overview of the QTN-based UQ pipeline used in our work. The procedure begins with computing the quantum correlation matrix (QCM) from TS probabilities and extracting its null-space vectors. These are used to construct the local Hamiltonian $\widehat{\underline{H}}$, whose eigen-modes provide the foundation for spectral and perturbation analysis. First-order perturbation corrections yield uncertainty feature vectors, which are organized into a tensor network representation. Finally, sampling across modes produces uncertainty bounds on probability amplitudes, enabling uncertainty quantification of semantic entropy.

Such a perturbation-based formulation of uncertainty cannot be obtained from SE because it lacks a corresponding RKHS-based representation.

Although SE achieves marginally better separability in AUROC due to its wider dynamic range, the empirical performance gap relative to SRE is small. In contrast, the methodological advantages facilitated by SRE—namely, its direct alignment with RKHS theory, its compatibility with QTN-based spectral analysis, and its ability to support principled perturbation-based uncertainty quantification—are substantially more consequential. For these reasons, SRE constitutes the most appropriate entropy measure for the proposed framework.

### A.4 RÉNYI ENTROPY AND KME OF DATA PDF

To provide the essentials of the work in Vipulananthan et al. (2024), take the Rényi (quadratic) entropy $H(X) = -\log(\psi(X))$ of the continuous random variable $X$ with PDF $p(x)$ Principe (2010). Here, $\psi(X) = \int p^2(x)\,dx = \mathbb{E}[p(x)]$, where $\mathbb{E}[\cdot]$ is the expectation operator. With $N$ samples $\{x_i\}_{i=1}^N$ from $X$, the non-parametric Parzen Gaussian density estimator of the PDF $p(x)$ is $\widehat{p}(x) = \sum_{i=1}^N \kappa_\sigma(x; x_i)/N$, where $\kappa_\sigma(x; x_i) = (1/\sqrt{2\pi\sigma^2})\exp(-(x-x_i)^2/2\sigma^2)$ Parzen (1962). This turns out to be the (empirical) KME $\widehat{\psi}(x)$ of the data PDF in the RKHS determined by $\kappa_\sigma(\cdot; \cdot)$ (Aronszajn (1950); Schölkopf et al. (2015)), i.e., $\widehat{p}(x) = \widehat{\psi}(x)$. $H(X) = -\log(\psi(X))$ being an entropy measure, the scalar $\widehat{\psi}(X)$ is taken as a measure of uncertainty and the (empirical) KME $\widehat{\psi}(x)$ viewed as how it varies with $x$, the data amplitude values.

To get a more interpretable tool for UQ, Vipulananthan et al. (2024) seeks a finite QTN having local nearest neighbor random couplings whose Hamiltonian has this KME $\widehat{\psi}(x)$ as *one* of its eigen-modes. Suffice it to say that **(a)** a spin chain having $L$ spin particles can represent a sampled version $\widehat{\underline{\psi}} = \{\widehat{\psi}_j\}_{j=0}^{M-1}$, where $M = 2^L$, of the KME $\widehat{\psi}(x)$; **(b)** the Hamiltonian of the TN takes the form of a linear combination of a finite set $\widehat{\mathcal{H}} = \{\widehat{\underline{H}}_k\}_{k=0}^{T-1}$ of pairwise orthonormal (w.r.t. the Hilbert-Schmidt inner product) Hermitian operators, where the extent of local nearest neighbor influence determines $T$; and **(c)** $T$, in turn, determines whether the set $\widehat{\mathcal{H}}$ is 'rich' enough for a Hamiltonian of the type $\widehat{\underline{H}} = \sum_{k=0}^{T-1} w_k \widehat{\underline{H}}_k$ (i.e., $\widehat{\underline{H}} \in \mathrm{span}(\widehat{\mathcal{H}})$) to have $\widehat{\underline{\psi}}$ as an eigen-mode.

To identify such a Hamiltonian, the quantum correlation matrix (QCM) $\underline{M}_{\widehat{\psi}}(\widehat{\mathcal{H}})$ (Qi & Ranard (2019)) because $\widehat{\underline{\psi}}$ is an eigen-mode of $\widehat{\underline{H}}$ iff $\underline{w}$ belongs in the null space of the QCM. Here, $\underline{w} = \{w_k\}_{k=0}^{T-1}$ is the linear combination weight vector that generates $\widehat{\underline{H}} = \sum_{k=0}^{T-1} w_k \widehat{\underline{H}}_k$. Additional theoretical results and supporting lemmas related to the QCM formulation are provided in Appendix A.5.

With this Hamiltonian in hand, the well-established method of perturbation theory is applied to the associated time-independent Schrödinger equation. This yields first- and higher order 'corrections' to *all* the eigen-modes/energies (including the KME) when the underlying Hamiltonian undergoes a perturbation (Cohen-Tannoudji et al. (1977)). The higher order modes provide better discriminative resolution of those data regions for which less information is available (these correspond to the tails of the PDF) (Singh & Principe (2021)); sensitivity analysis and UQ can be carried out via the corrections to the eigen-modes. This is the UQ method in Vipulananthan et al. (2024) that allows one to survey aleatoric uncertainties locally and at different levels of resolution using algebraic and spectral properties of QTNs.

## A.5 SOME PROPERTIES OF THE QCM

In this section, we present a series of results related to the QCM that form the theoretical backbone of our work, discussed in Section 2.1.

Note that $\widehat{\mathcal{H}} = \{\underline{\widehat{H}}_i\}$ denotes a finite set $\{\underline{\widehat{H}}_i,\ i = 0, \ldots, T-1\}$ of pairwise orthonormal Hermitian operators. For convenience, we will use $(\widehat{\mathcal{H}})_{\underline{w}}$ to denote a (real-valued) linear combination of the operators in $\widehat{\mathcal{H}}$, i.e.,

$$(\widehat{\mathcal{H}})_{\underline{w}} = \sum_{i=0}^{T-1} w_i \underline{\widehat{H}}_i, \text{ where } \underline{w} = \{w_i\}_{i=0}^{T-1} \in \mathbb{R}^T. \tag{8}$$

**Lemma 1** *The QCM $\underline{M}_{\underline{v}}(\mathcal{H}) = \{(\underline{M}_{\underline{v}}(\mathcal{H}))_{ij}\}$ associated with $\widehat{\mathcal{H}}$ is real and symmetric.*

*Proof.* From Definition, we have

$$(\underline{M}_{\underline{v}}(\mathcal{H}))_{ji} = 0.5 \langle \{\underline{\widehat{H}}_j, \underline{\widehat{H}}_i\} \rangle_{\underline{v}} - \langle \underline{\widehat{H}}_j \rangle_{\underline{v}} \cdot \langle \underline{\widehat{H}}_i \rangle_{\underline{v}}$$
$$= 0.5 \langle \{\underline{\widehat{H}}_i, \underline{\widehat{H}}_j\} \rangle_{\underline{v}} - \langle \underline{\widehat{H}}_i \rangle_{\underline{v}} \cdot \langle \underline{\widehat{H}}_j \rangle_{\underline{v}} = (\underline{M}_{\underline{v}}(\mathcal{H}))_{ij},$$

because $\{\underline{\widehat{H}}_i, \underline{\widehat{H}}_j\} = \{\underline{\widehat{H}}_j, \underline{\widehat{H}}_i\}$ and both $\langle \underline{\widehat{H}}_i \rangle_{\underline{v}}$ and $\langle \underline{\widehat{H}}_j \rangle_{\underline{v}}$ are scalars. So, $\underline{M}$ is symmetric.

In addition,

$$(\underline{M}_{\underline{v}}(\mathcal{H}))_{ij}^* = 0.5 \langle \{\underline{\widehat{H}}_i, \underline{\widehat{H}}_j\} \rangle_{\underline{v}}^* - \langle \underline{\widehat{H}}_i \rangle_{\underline{v}}^* \cdot \langle \underline{\widehat{H}}_j \rangle_{\underline{v}}^*$$
$$= 0.5 \langle \underline{v} \mid \{\underline{\widehat{H}}_i, \underline{\widehat{H}}_j\} \mid \underline{v} \rangle^* - \langle \underline{v} \mid \underline{\widehat{H}}_i \mid \underline{v} \rangle^* \cdot \langle \underline{v} \mid \underline{\widehat{H}}_j \mid \underline{v} \rangle^*$$
$$= 0.5 \langle \underline{v} \mid \underline{\widehat{H}}_i \underline{\widehat{H}}_j \mid \underline{v} \rangle^* + 0.5 \langle \underline{v} \mid \underline{\widehat{H}}_j \underline{\widehat{H}}_i \mid \underline{v} \rangle^* - \langle \underline{v} \mid \underline{\widehat{H}}_i \mid \underline{v} \rangle^* \cdot \langle \underline{v} \mid \underline{\widehat{H}}_j \mid \underline{v} \rangle^*$$

But we note that

$$\langle \underline{v} \mid \underline{\widehat{H}}_i \underline{\widehat{H}}_j \mid \underline{v} \rangle^* = \langle \underline{\widehat{H}}_i \underline{v} \mid \underline{\widehat{H}}_j \underline{v} \rangle^* = \langle \underline{\widehat{H}}_j \underline{v} \mid \underline{\widehat{H}}_i \underline{v} \rangle = \langle \underline{v} \mid \underline{\widehat{H}}_j \underline{\widehat{H}}_i \mid \underline{v} \rangle;$$
$$\langle \underline{v} \mid \underline{\widehat{H}}_i \mid \underline{v} \rangle^* \quad = \langle \underline{v} \mid \underline{\widehat{H}}_i \underline{v} \rangle^* \quad = \langle \underline{\widehat{H}}_i \underline{v} \mid \underline{v} \rangle \quad = \langle \underline{v} \mid \underline{\widehat{H}}_i \mid \underline{v} \rangle.$$

In a similar manner, we can also show that $\langle \underline{v} \mid \underline{\widehat{H}}_j \underline{\widehat{H}}_i \mid \underline{v} \rangle^* = \langle \underline{v} \mid \underline{\widehat{H}}_i \underline{\widehat{H}}_j \mid \underline{v} \rangle^*$ and $\langle \underline{v} \mid \underline{\widehat{H}}_j \mid \underline{v} \rangle^* = \langle \underline{v} \mid \underline{\widehat{H}}_j \mid \underline{v} \rangle$. When substituted into the expression above for $(\underline{M}_{\underline{v}}(\mathcal{H}))_{ij}^*$, we see that $(\underline{M}_{\underline{v}}(\mathcal{H}))_{ij}^* = (\underline{M}_{\underline{v}}(\mathcal{H}))_{ij}$. So, $\underline{M}$ is real. ∎

**Lemma 2** *The variance $\mathrm{Var}\,(\underline{\widehat{H}})_{\underline{\psi}} = \langle \underline{\widehat{H}}^2 \rangle_{\underline{\psi}} - |\langle \underline{\widehat{H}} \rangle_{\underline{\psi}}|^2$ of the Hermitian operator $\underline{\widehat{H}}$ w.r.t. an arbitrary normalized vector $\underline{\psi}$, $\langle \underline{\psi} \mid \underline{\psi} \rangle = 1$, satisfies*

$$\mathrm{Var}\,(\underline{\widehat{H}})_{\underline{\psi}} = \langle \underline{\widehat{H}}^2 \rangle_{\underline{\psi}} - |\langle \underline{\widehat{H}} \rangle_{\underline{\psi}}|^2 \geq 0;$$

*the equality holds iff $\underline{\psi}$ is an eigen-mode of $\underline{\widehat{H}}$.*

*Proof.* For arbitrary $\alpha \in \mathbb{C}$ and $\underline{\psi}$, we know that

$$0 \leq \langle \underline{\widehat{H}}\,\underline{\psi} - \alpha\,\underline{\psi},\ \underline{\widehat{H}}\,\underline{\psi} - \alpha\,\underline{\psi} \rangle.$$

Furthermore, the equality holds true iff $\underline{\widehat{H}}\,\underline{\psi} - \alpha\,\underline{\psi} = 0$, i.e., iff $\underline{\psi}$ is an eigen-mode of $\underline{\widehat{H}}$.

So, let us consider the case when $\underline{\psi}$ is not an eigen-mode of $\underline{\widehat{H}}$ so that

$$0 < \langle \underline{\widehat{H}}\,\underline{\psi} - \alpha\,\underline{\psi}, \underline{\widehat{H}}\,\underline{\psi} - \alpha\,\underline{\psi} \rangle = \langle \underline{\widehat{H}}\,\underline{\psi} \mid \underline{\widehat{H}}\,\underline{\psi} \rangle - \langle \underline{\widehat{H}}\,\underline{\psi} \mid \alpha\,\underline{\psi} \rangle - \langle \alpha\,\underline{\psi} \mid \underline{\widehat{H}}\,\underline{\psi} \rangle + \langle \alpha\,\underline{\psi} \mid \alpha\,\underline{\psi} \rangle$$
$$= \langle \underline{\widehat{H}}\,\underline{\psi} \mid \underline{\widehat{H}}\,\underline{\psi} \rangle - \alpha\,\langle \underline{\widehat{H}}\,\underline{\psi} \mid \underline{\psi} \rangle - \alpha^*\langle \underline{\psi} \mid \underline{\widehat{H}}\,\underline{\psi} \rangle + |\alpha|^2\,\langle \underline{\psi} \mid \underline{\psi} \rangle.$$

Since $\alpha$ is arbitrary, select

$$\alpha = \langle \underline{\psi} \mid \underline{\widehat{H}}\,\underline{\psi} \rangle \implies \alpha^* = \langle \underline{\widehat{H}}\,\underline{\psi} \mid \underline{\psi} \rangle.$$

Substitute into the above inequality:

$$0 < \langle \underline{\psi} \mid \underline{\widehat{H}}^2 \mid \underline{\psi} \rangle - |\langle \underline{\psi} \mid \underline{\widehat{H}}\,\underline{\psi} \rangle|^2 - |\langle \underline{\psi} \mid \underline{\widehat{H}}\,\underline{\psi} \rangle|^2 + |\langle \underline{\psi} \mid \underline{\widehat{H}}\,\underline{\psi} \rangle|^2$$
$$= \langle \underline{\psi} \mid \underline{\widehat{H}}^2 \mid \underline{\psi} \rangle - |\langle \underline{\psi} \mid \underline{\widehat{H}} \mid \underline{\psi} \rangle|^2 = \langle \underline{\widehat{H}}^2 \rangle_{\underline{\psi}} - |\langle \underline{\widehat{H}} \rangle_{\underline{\psi}}|^2.$$

This establishes the claim. ∎

The following result follows immediately:

**Corollary 3** *Given the Hermitian operator $\underline{\widehat{H}}$ and the normalized vector $\underline{\psi}$, $\langle \underline{\psi} \mid \underline{\psi} \rangle = 1$, $\mathrm{Var}\,(\underline{\widehat{H}})_{\underline{\psi}} = 0$ iff $\underline{\psi}$ is an eigen-mode of $\underline{\widehat{H}}$.*

**Corollary 4** *Suppose, for some arbitrary real-valued vector $\underline{w} \in \{w_i\} \in \mathbb{R}^T$, $\underline{\widehat{H}} = (\widehat{\mathcal{H}})_{\underline{w}}$. Then the following are true:*

*(i) For arbitrary $\underline{v} \in \mathbb{R}^T$, $\underline{M}_{\underline{v}}(\mathcal{H})$ is p.s.d.*

*(ii) $\underline{w} \in \mathscr{N}(\underline{M}_{\underline{v}}(\mathcal{H}))$, the null space of the QCM $\underline{M}_{\underline{v}}(\mathcal{H})$ associated with $\widehat{\mathcal{H}}$ w.r.t $\underline{v}$ iff $\underline{v}$, $\langle \underline{v} \mid \underline{v} \rangle = 1$, is an eigen-mode of $\underline{\widehat{H}}$.*

*Proof.*

(i) This follows directly from Lemma 2 and Qi & Ranard (2019).

(ii) First, suppose $\underline{w} \in \mathscr{N}(\underline{M}_{\underline{v}}(\mathcal{H}))$ so that

$$\underline{M}_{\underline{v}}(\mathcal{H})\,\underline{w} = 0 \implies \langle \underline{M}_{\underline{v}}(\mathcal{H}) \rangle_{\underline{w}} = \underline{w}^T \underline{M}_{\underline{v}}(\mathcal{H})\,\underline{w} = 0.$$

Then, $\mathrm{Var}\,(\underline{\widehat{H}})_{\underline{v}} = \langle \underline{M}_{\underline{v}}(\mathcal{H}) \rangle_{\underline{w}} = 0$ (Qi & Ranard (2019)). Corollary 3 then implies that $\underline{v}$ is an eigen-mode of $\underline{\widehat{H}}$.

Conversely, suppose $\langle \underline{M}_{\underline{v}}(\mathcal{H}) \rangle_{\underline{w}} = \underline{w}^T \underline{M}_{\underline{v}}(\mathcal{H})\,\underline{w} = 0$. We know from Lemma 1 that $\underline{M}_{\underline{v}}(\mathcal{H})$ is real and symmetric. Let its SVD be

$$\underline{M}_{\underline{v}}(\mathcal{H}) = \underline{U}\,\underline{\Sigma}\,\underline{U}^T,$$

where $\underline{U} \in \mathbb{R}^{T \times T}$ is unitary and $\underline{\Sigma} \in \mathbb{R}^{T \times T}$ is diagonal with its diagonal entries being the non-negative singular values of $\underline{M}_{\underline{v}}(\mathcal{H})$. Let $\underline{M}_{\underline{v}}(\mathcal{H})^{1/2} = \underline{\Sigma}^{1/2}\underline{U}^T$. Then,

$$\underline{w}^T \underline{M}_{\underline{v}}(\mathcal{H})\,\underline{w} = \underline{w}^T \underline{M}_{\underline{v}}(\mathcal{H})^{1/2^T} \underline{M}_{\underline{v}}(\mathcal{H})^{1/2}\underline{w} = \langle \underline{M}_{\underline{v}}(\mathcal{H})^{1/2}\underline{w}, \underline{M}_{\underline{v}}(\mathcal{H})^{1/2}\underline{w} \rangle = 0,$$

meaning that we must have

$$\underline{M}_{\underline{v}}(\mathcal{H})^{1/2}\underline{w} = 0 \implies \underline{M}_{\underline{v}}(\mathcal{H})\,\underline{w} = 0,$$

i.e., $\underline{w} \in \mathscr{N}(\underline{M}_{\underline{v}}(\mathcal{H}))$. ∎

## A.6 ON THE NULL SPACE OF THE QCM: THE ZERO DIMENSION CASE

We now study a special case where the QCM has full rank (zero-dimensional null space) and investigate how small perturbations can introduce a one-dimensional null space. Understanding this

transition is important because the emergence of a nontrivial null space corresponds to identifying new uncertainty directions, which are critical for robust decision-making in our framework.

Consider the case in the QCM, when the dimension of the null space of the QCM $\underline{M}_{\psi^\sharp}(\mathcal{H})$ is zero and we 'perturb' $\underline{M}_{\psi^\sharp}(\mathcal{H})$ to $\underline{M}_{\psi^\sharp}(\mathcal{H}')$ as

$$\underline{M}_{\psi^\sharp}(\mathcal{H}') = \underline{M}_{\psi^\sharp}(\mathcal{H}) - \delta\underline{M} \tag{9}$$

s.t. $\dim\left(\mathcal{N}\left(\underline{M}_{\psi^\sharp}(\mathcal{H}')\right)\right) = 1$.

Henceforth in this section, for convenience, we will denote $\underline{M}_{\psi^\sharp}(\mathcal{H})$ by $\underline{M}$ and $\underline{M}_{\psi^\sharp}(\mathcal{H}')$ by $\underline{M}'$.

Recall that $\{\underline{w}_n, \mu_n\}$, $n \in 0, 1, \ldots, T-1$, denote the eigen-pairs of $\underline{M}$ ordered as $0 < \mu_0 \le \mu_1 \le \cdots \le \mu_{T-1}$.

We next investigate the stability of the QCM under small perturbations. Specifically, we examine how the eigen-modes/energies of the QCM change when a small perturbation is introduced, particularly in the context where the original QCM has full rank (no null space) and perturbations induce a null direction. We use classical matrix perturbation theory to derive first-order approximations and cosine similarity results.

Let $\{\underline{w}'_n, \mu'_n\}$, $n \in 0, 1, \ldots, T-1$, denote the eigen-pairs of $\underline{M}'$ ordered as $0 = \mu'_0 < \mu'_1 \le \cdots \le \mu'_{T-1}$. Let us now view the eigen-pairs $\{\underline{w}_n, \lambda_n\}$ of $\underline{M}$ as the 'perturbed' versions of the eigen-pairs $\{\underline{w}'_n, \lambda'_n\}$ of $\underline{M}'$, so that we may apply perturbation theory to $\underline{M}'$. For this purpose, as is customary in perturbation theory, let

$$\underline{M} = \underline{M}' + \epsilon \cdot \delta\underline{M}, \tag{10}$$

where $\epsilon > 0$ is simply a place-holder. The eigen-pairs of $\underline{M}$ can then be expressed as

$$\{\underline{w}_n, \lambda_n\} = \left\{ \sum_{k=0}^{\infty} \epsilon^k \underline{w}_n^{(k)}, \sum_{\ell=0}^{\infty} \epsilon^\ell \lambda_n^{(\ell)} \right\}, \tag{11}$$

where $\{w_n^{(0)}, \lambda_n^{(0)}\} \triangleq \{w'_n, \lambda'_n\}$, $\forall n \ge 0$. W.l.o.g. we also assume that $\langle w_n^{(0)} \mid \underline{w}_n^{(k)} \rangle = 0$, $\forall k \ge 1$, $\forall n \ge 0$.

Now we can express the eigen-pair relationship

$$\underline{M} \mid \underline{w}_n \rangle = \lambda_n \mid \underline{w}_n \rangle \tag{12}$$

as

$$(\underline{M}' + \epsilon \cdot \delta\underline{M}) \mid \sum_{k=0}^{\infty} \epsilon^k \underline{w}_n^{(k)} \rangle = \sum_{\ell=0}^{\infty} \epsilon^\ell \lambda_n^{(\ell)} \mid \sum_{k=0}^{\infty} \epsilon^k \underline{w}_n^{(k)} \rangle, \tag{13}$$

or equivalently,

$$\underline{M}' \sum_{k=0}^{\infty} \epsilon^k \mid \underline{w}_n^{(k)} \rangle + \delta\underline{M} \sum_{k=1}^{\infty} \epsilon^k \mid \underline{w}_n^{(k)} \rangle = \sum_{k=0}^{\infty} \epsilon^k \sum_{\ell=0}^{k} \lambda_n^{(k-\ell)} \mid \underline{w}_n^{(\ell)} \rangle. \tag{14}$$

Gather similar powers of $\epsilon$: for $k \ge 1$,

$$(\underline{M}' - \lambda_n^{(0)}) \mid \underline{w}_n^{(k)} \rangle = \sum_{\ell=0}^{k-1} \lambda_n^{(k-\ell)} \mid \underline{w}_n^{(\ell)} \rangle - \delta\underline{M} \mid \underline{w}_n^{(k-1)} \rangle. \tag{15}$$

Note that, with $k = 0$, we get $(\underline{M}' - \lambda_n^{(0)}) \mid \underline{w}_n^{(0)} \rangle = 0$, which is of course trivially true.

Push the bra $\langle \underline{w}_m^{(0)} \mid$: for $k \ge 1$,

$$(\lambda_m^{(0)} - \lambda_n^{(0)})\langle \underline{w}_m^{(0)} \mid \underline{w}_n^{(k)} \rangle = \sum_{\ell=0}^{k-1} \lambda_n^{(k-\ell)}\langle \underline{w}_m^{(0)} \mid \underline{w}_n^{(\ell)} \rangle - \langle \underline{w}_m^{(0)} \mid \delta\underline{M} \mid \underline{w}_n^{(k-1)} \rangle. \tag{16}$$

Put $m = n$ in equation 16 to get the order-$k$ correction for $\lambda_n^{(0)}$:

$$\lambda_n^{(k)} = \langle \underline{w}_n^{(0)} \mid \delta\underline{M} \mid \underline{w}_n^{(k-1)} \rangle, \ k \ge 1. \tag{17}$$

For $m \neq n$, put $k = 1$ in equation 16 to get

$$\langle \underline{w}_m^{(0)} \mid \underline{w}_n^{(1)} \rangle = \frac{\langle \underline{w}_m^{(0)} \mid \delta \underline{M} \mid \underline{w}_n^{(0)} \rangle}{\lambda_n^{(0)} - \lambda_m^{(0)}}, \tag{18}$$

where we assume that $\underline{M}^{(0)}$ is non-degenerate. Since $\{\underline{w}_n^{(0)}\}$ forms an ONB, we can get the order-1 correction for $\underline{w}_n^{(0)}$ as

$$\mid \underline{w}_n^{(1)} \rangle = \sum_{m=0}^{\infty} \mid \underline{w}_m^{(0)} \rangle \cdot \langle \underline{w}_m^{(0)} \mid \underline{w}_n^{(1)} \rangle = \sum_{\substack{m=0 \\ m \neq n}}^{\infty} \mid \underline{w}_m^{(0)} \rangle \cdot \frac{\langle \underline{w}_m^{(0)} \mid \delta \underline{M} \mid \underline{w}_n^{(0)} \rangle}{\lambda_n^{(0)} - \lambda_m^{(0)}}. \tag{19}$$

Notice that, since $\langle \underline{w}_n^{(0)} \mid \underline{w}_n^{(1)} \rangle = 0$, we have

$$\|\underline{w}_n^{(0)} + \underline{w}_n^{(1)}\|^2 = \langle \underline{w}_n^{(0)} \mid \underline{w}_n^{(0)} \rangle + \langle \underline{w}_n^{(1)} \mid \underline{w}_n^{(1)} \rangle = 1 + \|\underline{w}_n^{(1)}\|^2, \tag{20}$$

because $\underline{w}_n^{(0)}$ is already normalized. Therefore, denoting by $\widehat{\underline{w}}_n^{(0)}$ the normalized order-1 corrected $\underline{w}_n^{(0)}$, we have

$$\widehat{\underline{w}}_n^{(0)} = \frac{\underline{w}_n^{(0)} + \underline{w}_n^{(1)}}{(1 + \|\underline{w}_n^{(1)}\|^2)^{1/2}}, \tag{21}$$

where

$$\begin{aligned}
\|\underline{w}_n^{(1)}\|^2 &= \langle \underline{w}_n^{(1)} \mid \underline{w}_n^{(1)} \rangle \\
&= \sum_{\ell \neq n} \sum_{m \neq n} \frac{\langle \underline{w}_m^{(0)} \mid \delta \underline{M} \mid \underline{w}_n^{(0)} \rangle}{\lambda_n^{(0)} - \lambda_m^{(0)}} \cdot \frac{\langle \underline{w}_\ell^{(0)} \mid \delta \underline{M} \mid \underline{w}_n^{(0)} \rangle}{\lambda_n^{(0)} - \lambda_\ell^{(0)}} \cdot \langle \underline{w}_\ell^{(0)} \mid \underline{w}_m^{(0)} \rangle \\
&= \sum_{m \neq n} \left( \frac{\langle \underline{w}_m^{(0)} \mid \delta \underline{M} \mid \underline{w}_n^{(0)} \rangle}{\lambda_n^{(0)} - \lambda_m^{(0)}} \right)^2.
\end{aligned} \tag{22}$$

Thus,

$$\begin{aligned}
\langle \underline{w}_n^{(0)} \mid \widehat{\underline{w}}_n^{(0)} \rangle &= \frac{\langle \underline{w}_n^{(0)} \mid \underline{w}_n^{(0)} \rangle + \langle \underline{w}_n^{(0)} \mid \underline{w}_n^{(1)} \rangle}{(1 + \|\underline{w}_n^{(1)}\|^2)^{1/2}} \\
&= \frac{1}{(1 + \|\underline{w}_n^{(1)}\|^2)^{1/2}} \geq 1 - \frac{1}{2} \|\underline{w}_n^{(1)}\|^2,
\end{aligned} \tag{23}$$

where we used the Taylor series expansion

$$\frac{1}{(1+x)^{1/2}} = 1 - \frac{1}{2} x + \frac{1}{2!} \frac{3}{4} (1+c)^{-5/2} x^2, \ x > 0, \tag{24}$$

for some $c \in (0, x)$, which in turn implies that

$$\frac{1}{(1+x)^{1/2}} \geq 1 - \frac{1}{2} x, \tag{25}$$

because $(1+c)^{-5/2} > 0, \ c \in (0, x)$.

In summary, we have

$$\langle \underline{w}_n^{(0)} \mid \widehat{\underline{w}}_n^{(0)} \rangle \geq 1 - \frac{1}{2} \sum_{m \neq n} \left( \frac{\langle \underline{w}_m^{(0)} \mid \delta \underline{M} \mid \underline{w}_n^{(0)} \rangle}{\lambda_n^{(0)} - \lambda_m^{(0)}} \right)^2. \tag{26}$$

We know that $\{\underline{w}_0^{(0)}, \lambda_0^{(0)}\} = \{\underline{w}_0', \lambda_0' = 0\}$ is the 'smallest' eigen-pair of $\underline{M}'$. So, putting $n = 0$, we have

$$\langle \underline{w}_0^{(0)} \mid \widehat{\underline{w}}_0^{(0)} \rangle \geq 1 - \frac{1}{2} \sum_{m \neq 0} \left( \frac{\langle \underline{w}_m^{(0)} \mid \delta \underline{M} \mid \underline{w}_0^{(0)} \rangle}{\lambda_m^{(0)}} \right)^2. \tag{27}$$

To conclude, this section shows that introducing a small perturbation to a full-rank QCM systematically creates a well-defined null space, corresponding to an emergent low-variance direction in the QTN. The perturbation analysis, grounded in classical matrix perturbation theory, quantifies how close the perturbed eigen-mode remains to the original eigen-structure through a first-order approximation. This understanding is critical in practice, as it ensures that small modeling errors or noise do not destabilize the uncertainty quantification pipeline and that the emergence of low-uncertainty directions remains theoretically sound and controllable.

## B  PROMPT TEMPLATES

**Phrase length Model Prompting and Answer Selection**    The prompt template used for all datasets to generate LLM responses is

> *Answer the following question as briefly as possible:* {question}
> *Answer:*

**Sentence length Model Prompting and Answer Selection**    The prompt template used for all datasets to generate LLM responses is

> *Answer the following question in a single brief but complete sentence:* {question}
> *Answer:*

**Entailment Estimation/Semantic Clustering**    To detect entailment we utilized the following prompt template:

> *We are evaluating answers to the question:* {question}
>
> *Here are two possible answers:*
> Possible Answer 1: {text1}
> Possible Answer 2: {text2}
>
> *Does Possible Answer 1 semantically entail Possible Answer 2?*
> *Respond with:* entailment, contradiction, or neutral.

## C  EXTENDED EVALUATION OF HALLUCINATION DETECTION ACROSS DIVERSE SETTINGS

Due to page limitations, we reported the results summary in the main paper. In this section, we present in depth experimental results to validate that our confabulation detection framework works across diverse datasets. Building on the example analysis in Section 4, we extend the evaluation to TriviaQA, NQ, SVAMP, and SQuAD across multiple quantization levels, thereby encompassing a broader spectrum of reasoning complexities and linguistic variations. In addition, we break down results for both phrase-level and sentence-level generations, via prompts discussed in Section B) to examine the impact of output length on detection robustness. For each dataset, we consistently apply the eight LLM models (Mistral-7B-v0.1, Mistral-7B-instruct-v0.3, Falcon-rw-1b, LLaMA-3.2-1b, LLaMA-2-13b-chat, LLaMA-2-7b-chat, LLaMA-2-13b, and LLaMA-2-7b) and maintain the original entailment-based clustering and uncertainty quantification pipelines without modification. This controlled experimental design ensures that any observed performance trends can be attributed to dataset characteristics rather than methodological changes. The results demonstrate that our framework generalizes effectively across heterogeneous tasks, reinforcing its applicability to varied real-world deployment scenarios.

### C.1  AUROC PERFORMANCE ACROSS VARIOUS PRECISION MODELS AND DATASETS

To more rigorously validate the robustness of our confabulation detection framework, we extend the evaluation to sentence-level and phrase-level generations under different quantization precision (16-bit, 8-bit, and 4-bit). This setup allows us to examine how model compression impacts uncertainty estimation and confabulation detection performance, when sentence length answers as well as short answers are generated. We consistently apply the set of LLMs and datasets as in the main paper, thereby ensuring that observed variations are attributable solely to quantization effects rather than changes in methodology. Importantly, quantization is a necessary step for deploying large models

in resource-constrained environments, yet hallucination behavior under quantized models has been largely overlooked in prior work. By explicitly studying confabulation detection in this setting and validating our approach against the SOTA methods discussed in Section 3, we address a critical gap in the literature and highlight the practical importance of uncertainty-aware methods for LLMs.

### C.1.1 SENTENCE LENGTH OUTPUT PERFORMANCE.

Figs. 6, 7, and 8 illustrate the AUROC performance at 16-bit, 8-bit, and 4-bit precision, respectively, for six different hallucination quantification methods across various LLMs and four datasets, using sentence-length outputs. In total, we conducted **52 experiments** spanning all quantization levels, providing a comprehensive basis for sentence length analysis. The experimental results reveal several notable trends and insights:

1. Methods based on semantic information—such as SRE with UQ, SE, and DSE—consistently outperform baselines based on raw token sequence probabilities ($p$(True)) and embedding regression across all datasets. These results confirm effectiveness of semantic diversity rather than raw output probabilities in detecting confabulations.
2. Across all quantization levels, our framework maintains competitive AUROC performance relative to baseline methods. Importantly, we observe that reductions in model precision (e.g., moving from 16-bit to 4-bit quantization) do not significantly degrade the reliability of our uncertainty-based detection.
3. When model precision is reduced, SOTA methods such as SE, NE have performance degradation, especially in TriviaQA and SQuAD datasets. This highlights the resilience of our approach to model compression, making it suitable for deployment in resource-constrained environments where lightweight LLMs are required.
4. Smaller models such as LLaMA 3.2-1B show slightly reduced AUROC compared to larger models like LLaMa-13B, revealing that model size and internal representational power are significant factors in uncertainty calibration.
5. We also observe a minor performance drop for the NQ dataset when using the LLaMA 3.1B model, which warrants further investigation in future work.

### C.1.2 PHRASE LENGTH OUTPUT PERFORMANCE ON INSTRUCT MODELS.

In this section, we extend the analysis to phrase-level generations, focusing on instruction-tuned LLMs, such as under multiple quantization precisions (16-bit, 8-bit, and 4-bit). Phrase-level evaluation captures shorter, context-sensitive completions, which are particularly relevant for instruction-following scenarios where answers are often concise. By applying the same uncertainty quantification pipeline across datasets and models, we ensure methodological consistency, allowing observed performance variations to be attributed to quantization and model adaptation effects. While quantization has become standard for enabling deployment of instruction-tuned LLMs in practical applications, its influence on hallucination behavior remains unexplored. Our experiments directly addresses this gap by examining confabulation detection under compressed, instruction-following settings.

Figs. 9, 10, and 11 present the AUROC scores for six hallucination quantification approaches across diverse datasets and instruction-tuned LLMs, under 16-bit, 8-bit, and 4-bit precision. Across all quantization levels, we performed a total of **24 experiments**, providing a robust foundation for phrase-level evaluation on instruction fine tuned models. The results reveal several consistent patterns and distinctive insights:

1. Semantic-based criteria such as SRE with UQ maximization, SE, and DSE—consistently outperform probability-based ($p$(True)) and embedding-based baselines, indicating that semantic variability is a reliable uncertainty signal even for shorter responses.
2. Across all quantization levels, our framework maintains competitive AUROC performance relative to baseline methods. Importantly, we observe that reductions in model precision (e.g., moving from 16-bit to 4-bit quantization) do not significantly degrade the reliability of our uncertainty-based detection
3. By contrast, prior SOTA baselines such as SE and NE show larger declines under reduced precision, particularly in more challenging datasets (e.g., NQ, SQuAD), underscoring the sensitivity of token-level entropy measures to compression.

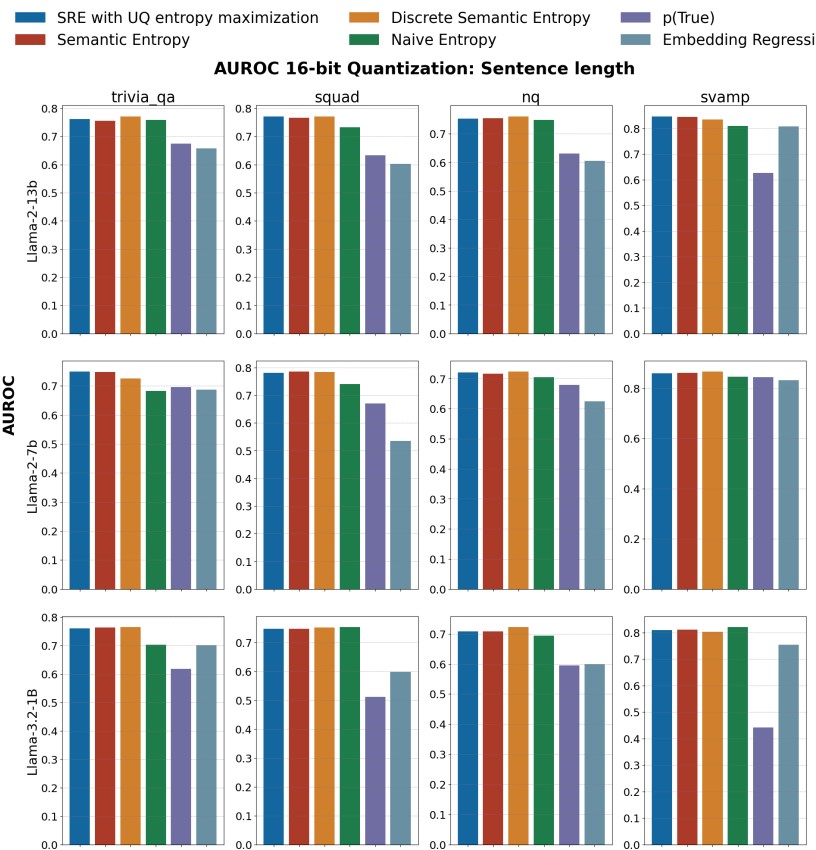

Figure 6: **summarizing 12 experimental** scenarios, AUROC scores for confabulation detection across three LLMs (LLaMA 2 13B, LLaMA 2 7B, LLaMA 3.2 1B) at 16 bit precision and four datasets (TriviaQA, SQuAD, NQ, SVAMP) for sentence length output. The performance of the proposed SRE with UQ is in par or even higher than SOTA methods.

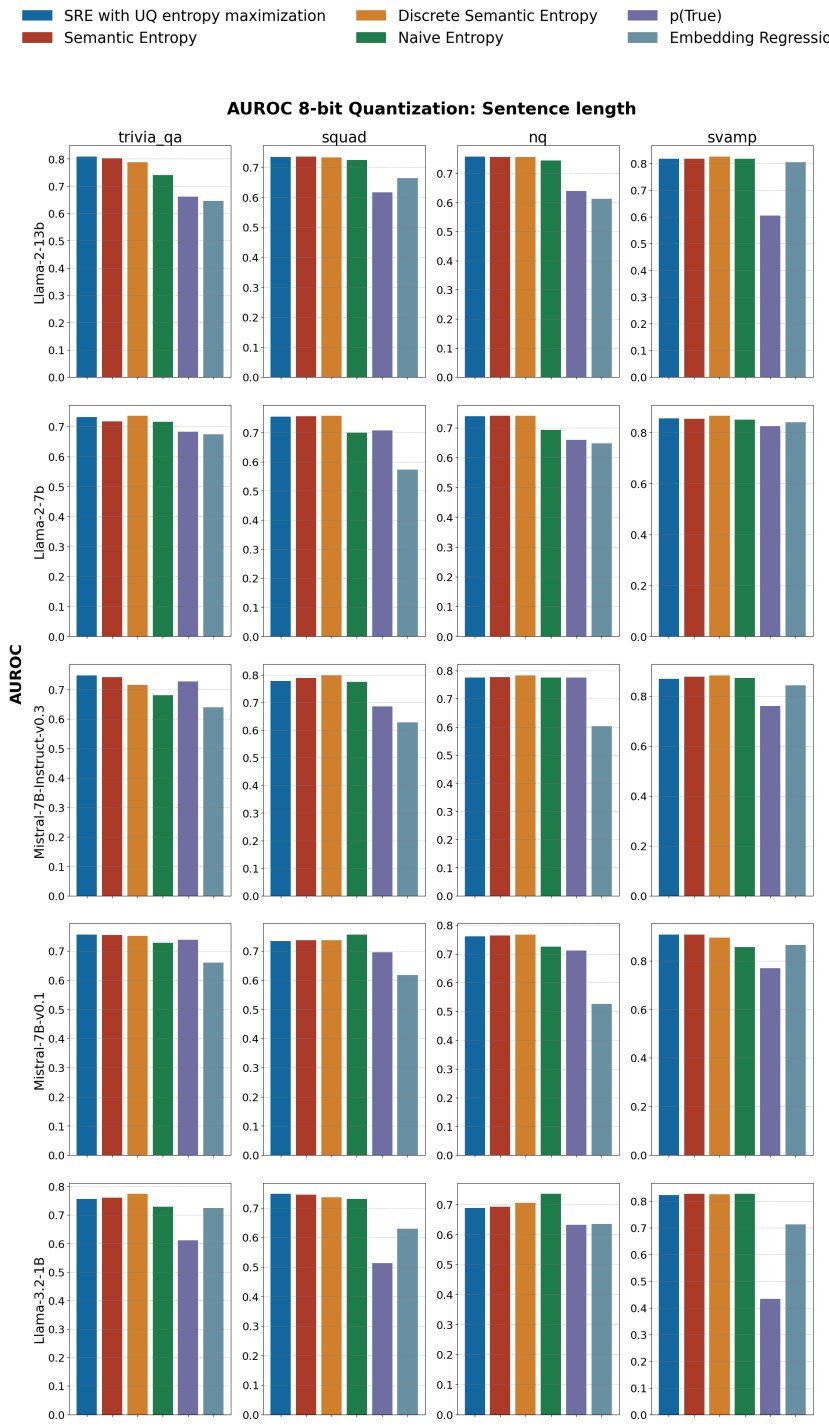

Figure 7: **summarizing 20 experimental** scenarios, AUROC scores for confabulation detection across five LLMs (LLaMA 2 13B, LLaMA 2 7B, LLaMA 3.2 1B, Mistral-7B, Mistral-7B-chat) at 8 bit precision and four datasets (TriviaQA, SQuAD, NQ, SVAMP) for sentence length output. The performance of the proposed SRE with UQ is in par or even higher than SOTA methods.

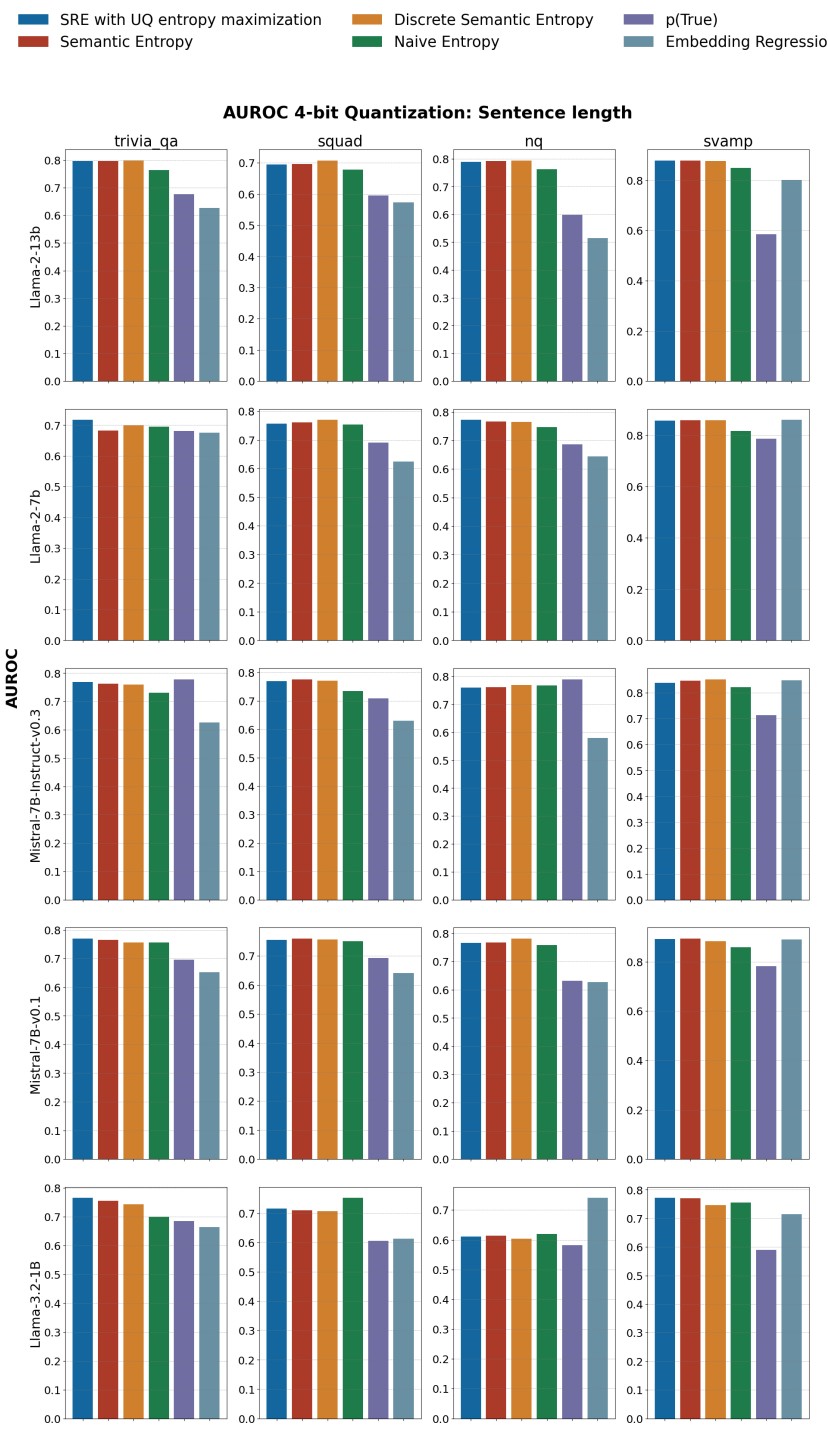

Figure 8: **summarizing 20 experimental** scenarios, AUROC scores for confabulation detection across five LLMs (LLaMA 2 13B, LLaMA 2 7B, LLaMA 3.2 1B, Mistral-7B, Mistral-7B-chat) at 4 bit precision and four datasets (TriviaQA, SQuAD, NQ, SVAMP) for sentence length output. The performance of the proposed SRE with UQ is in par or even higher than SOTA methods.

4. Instruction-tuned models provide better calibrated AUROC overall compared to their base versions, though a consistent gap remains between the smaller LLaMA-2-7B-chat and the larger LLaMA-2-13B-chat, highlighting the role of model scale in uncertainty reliability.

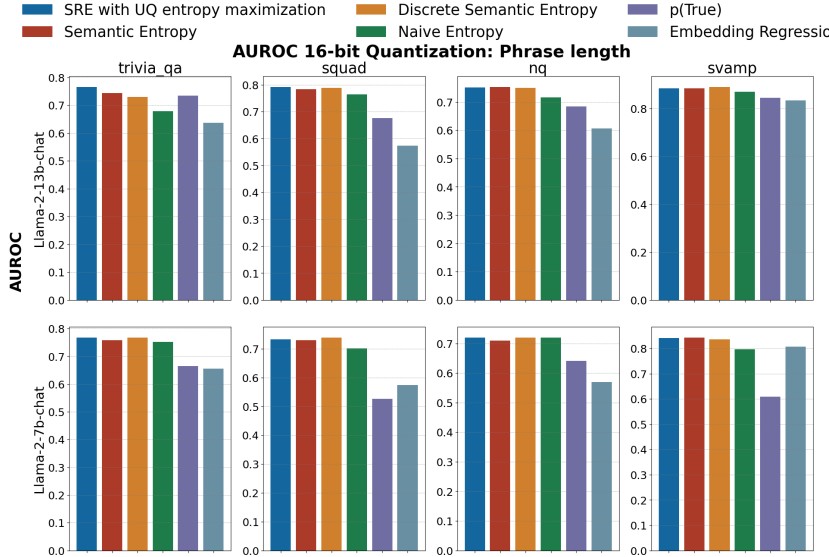

Figure 9: **summarizing 8 experimental** scenarios, AUROC scores for confabulation detection across two LLMs (LLaMA 2 7B chat, LLaMA 2 13B chat) at 16 bit precision and four datasets (TriviaQA, SQuAD, NQ, SVAMP). The performance of the proposed SRE with UQ is in par or even higher than SOTA methods.

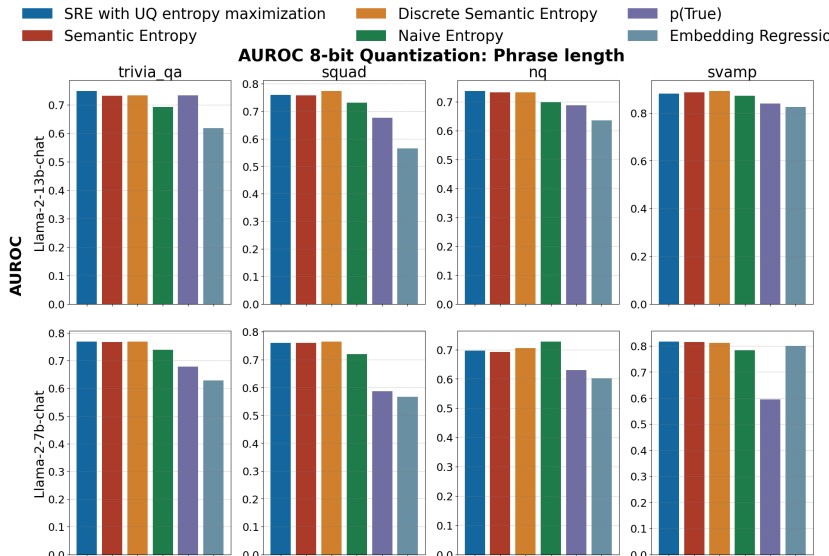

Figure 10: **summarizing 8 experimental** scenarios, AUROC scores for confabulation detection across four LLMs (Mistral-7B, Falcon-1B, LLaMA 3.2B, LLaMA 2 7B 4-bit) and four datasets (TriviaQA, SQuAD, NQ, SVAMP). The performance of the proposed semantic Rényi entropy is in par or even higher than SOTA methods.

### C.1.3 PHRASE LENGTH OUTPUT PERFORMANCE ON NON INSTRUCT MODELS.

In this section, we analyze phrase-level generations from non-instruction-tuned LLMs under varying quantization levels (16-bit, 8-bit, and 4-bit). Phrase-level completions in base models are especially

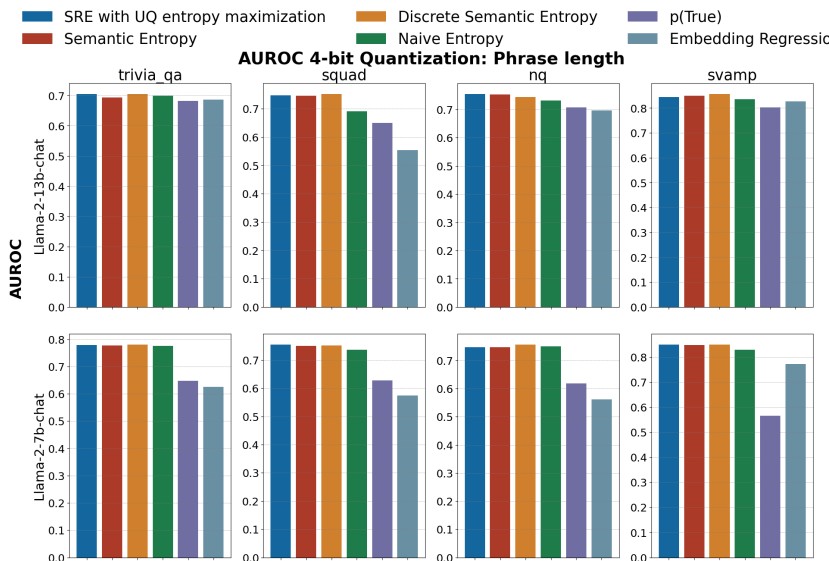

Figure 11: **summarizing 8 experimental** scenarios, AUROC scores for confabulation detection across four LLMs (Mistral-7B, Falcon-1B, LLaMA 3.2B, LLaMA 2 7B 4-bit) and four datasets (TriviaQA, SQuAD, NQ, SVAMP). The performance of the proposed semantic Rényi entropy is in par or even higher than SOTA methods.

prone to drifting and incoherent continuations because, unlike instruction-tuned models, they lack explicit task alignment. This makes them a critical yet underexplored regime for studying hallucination behavior—particularly when combined with quantization, which is essential for deploying large models in constrained environments. Prior SOTA methods have largely overlooked this intersection, focusing either on sentence-level or instruction-following setups, leaving open questions about uncertainty reliability in compressed base models. Our study directly addresses this gap by evaluating uncertainty-aware hallucination detection in non-instruction-tuned, phrase-level settings.

Figs. 12, 13, and 14 summarize AUROC performance across six uncertainty quantification methods, four datasets, and multiple base LLMs under 16-bit, 8-bit, and 4-bit quantization. In total, we conducted **40 experiments**, enabling a systematic comparison of quantization and uncertainty estimation in this overlooked regime. The results reveal the following consistent findings:

1. Semantic-based methods dominate: Across datasets and precisions, SRE with UQ, SE, and DSE consistently outperform $p(\text{True})$ and embedding regression, confirming that semantic diversity remains a strong uncertainty signal even in base models with less task alignment.
2. Across all quantization levels, our framework maintains competitive AUROC performance relative to baseline methods. Importantly, we observe that reductions in model precision (e.g., moving from 16-bit to 4-bit quantization) do not significantly degrade the reliability of our uncertainty-based detection
3. Token-level entropy baselines (NE) suffer performance drops—most evident in NQ and SQuAD—highlighting their fragility under quantization and emphasizing the robustness of semantic-based signals.
4. Larger base models (e.g., LLaMA-2-13B) consistently achieve higher AUROC than smaller ones (e.g., LLaMA-3.2-1B, Falcon-RW-1B). This indicates that representational richness is particularly crucial for phrase-level uncertainty calibration in non-instruct settings.

## C.2 RAC PERFORMANCE ACROSS VARIOUS PRECISION MODELS AND DATASETS

To more rigorously evaluate the robustness of our confabulation detection framework, we complement the AUROC analysis (Appendix C.1) with RAC and their area-under-curve summary, AURAC, across multiple quantization precisions (16-bit, 8-bit, and 4-bit). While AUROC measures separability under idealized conditions, RAC and AURAC provide a stricter evaluation by quantifying how reliably uncertainty estimates improve accuracy under progressive rejection. Given that quantization is now standard for deploying LLMs in resource-constrained environments, understanding its impact on

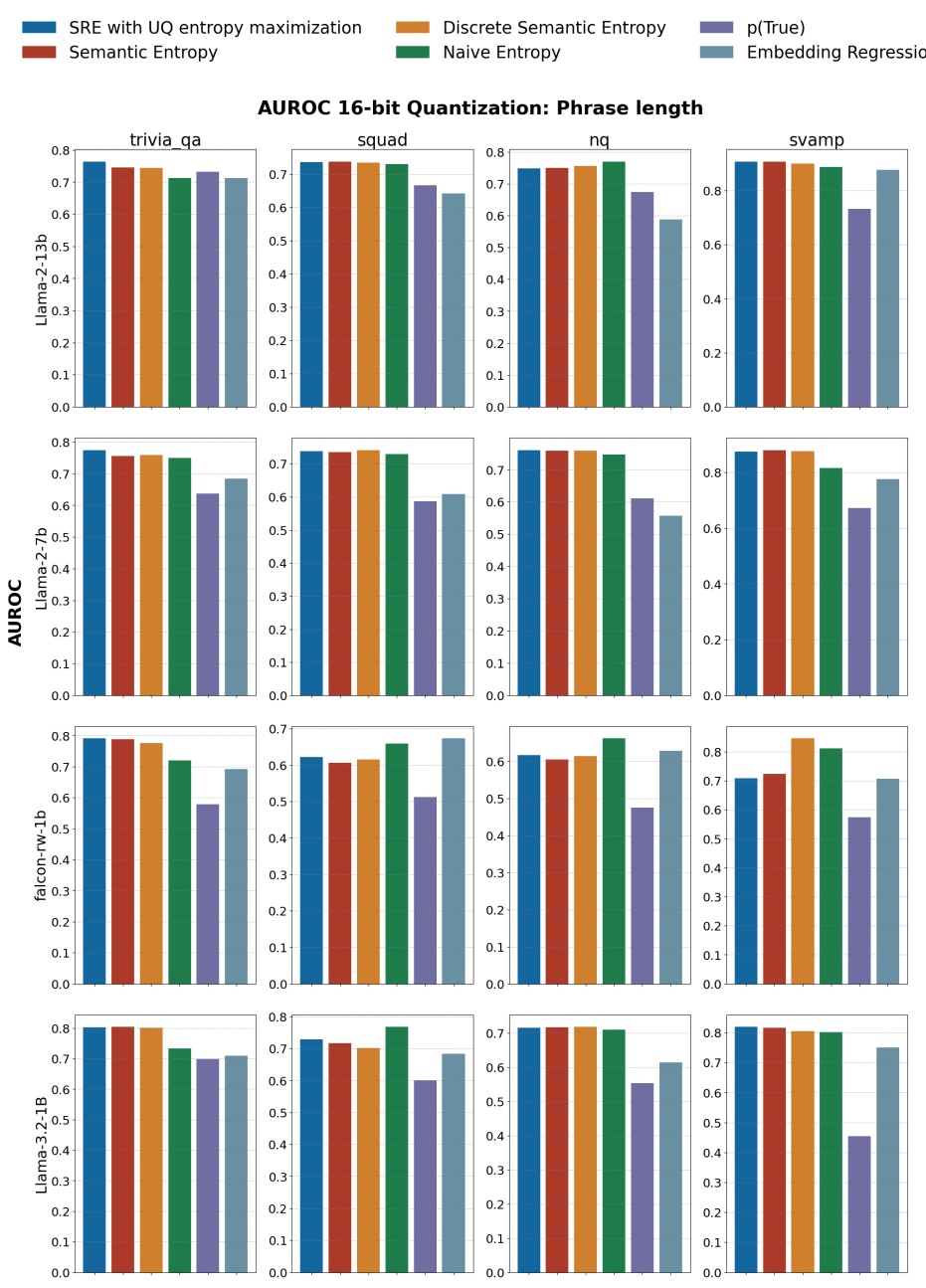

Figure 12: **summarizing 16 experimental** scenarios, AUROC scores for confabulation detection across four LLMs (Falcon-1B, LLaMA 3.2B, LLaMA 2 7B, LLaMA 2 13B) and four datasets (TriviaQA, SQuAD, NQ, SVAMP). The performance of the proposed SRE with UQ is in par or even higher than SOTA methods.

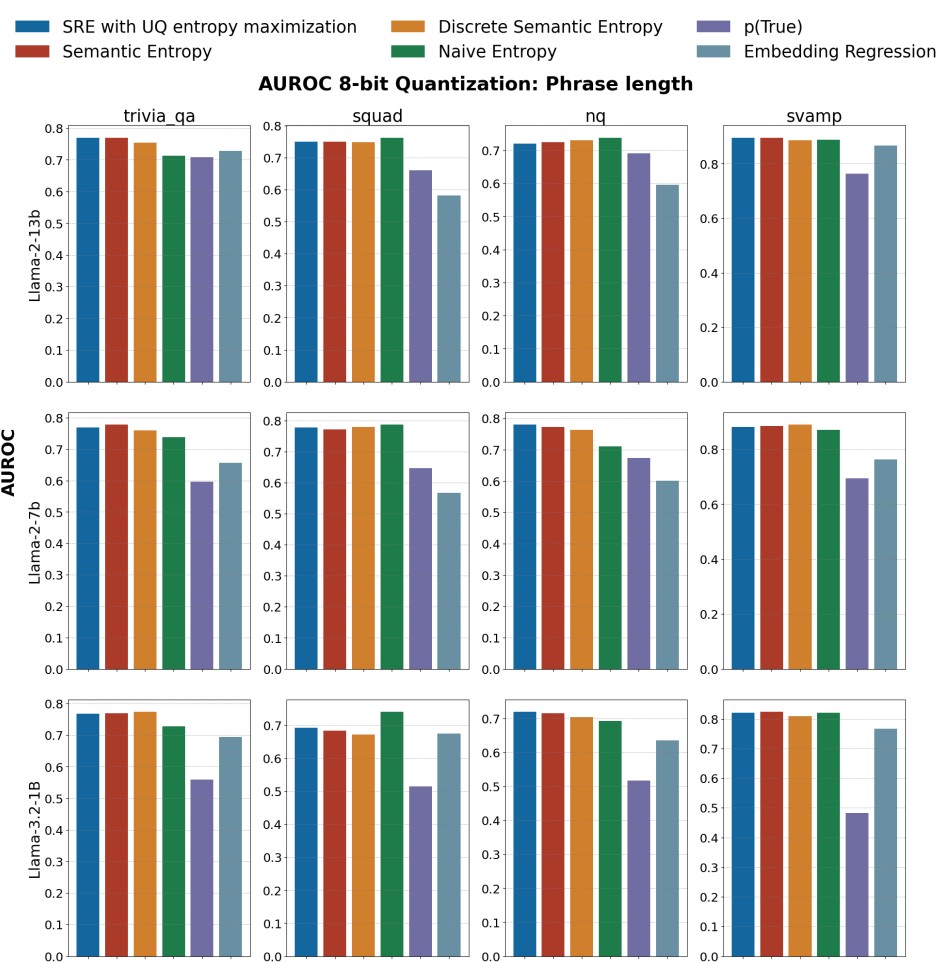

Figure 13: **summarizing 12 experimental** scenarios, AUROC scores for confabulation detection across three LLMs (LLaMA 3.2B, LLaMA 2 7B, LLaMA 2 13B) and four datasets (TriviaQA, SQuAD, NQ, SVAMP). The performance of the proposed SRE with UQ is in par or even higher than SOTA methods.

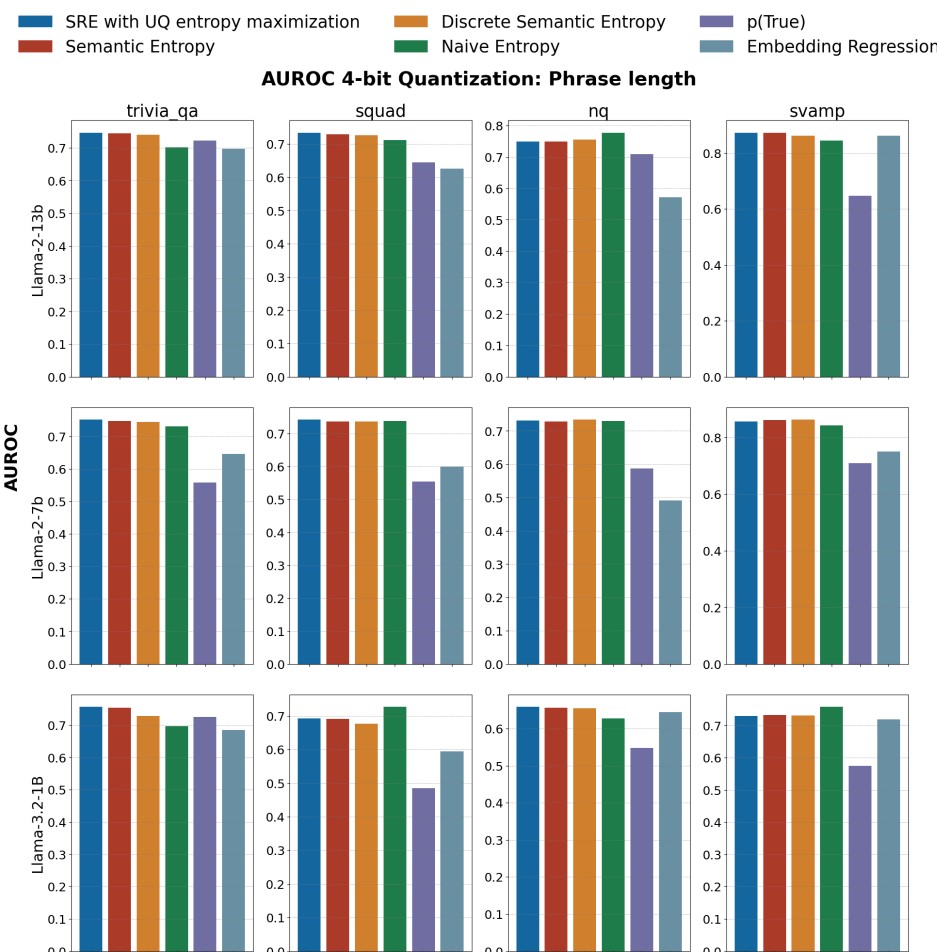

Figure 14: **summarizing 12 experimental** scenarios, AUROC scores for confabulation detection across three LLMs (LLaMA 3.2B, LLaMA 2 7B, LLaMA 2 13B) and four datasets (TriviaQA, SQuAD, NQ, SVAMP). The performance of the proposed SRE with UQ is in par or even higher than SOTA methods.

rejection behavior remains a critical but largely overlooked dimension. By systematically analyzing RACs and AURAC under quantization, we disentangle the effects of compression and model scale, ensuring methodological consistency via identical clustering-based uncertainty pipelines. In this setting, steeper RAC slopes indicate stronger capacity to prioritize reliable generations while filtering confabulations.

Expanding on the AURAC summary results reported in Section 4.2, our findings establish that the proposed SRE with UQ not only surpasses prior methods in robustness but also enables fine-grained and safe filtering of LLM outputs—demonstrating both higher accuracy and greater deployability than existing SOTA uncertainty estimators.

### C.2.1 RAC PERFORMANCE: SENTENCE LENGTH

Figs. 15, 16, and 17 present RAC curves at 16-bit, 8-bit, and 4-bit precision, respectively, for six hallucination quantification methods across multiple LLMs and four datasets under sentence-length generation. In total, we conducted **52 experiments** spanning all quantization levels, providing a comprehensive basis for sentence length analysis, drawing parallel with its AUROC counterpart evaluation (Appendix C.1.1). Several notable patterns emerge:

1. The proposed SRE with UQ consistently yields the steepest accuracy gains as rejection increases, demonstrating superior ability to prioritize reliable outputs and suppress confabulations. Other semantic-based criteria (SE and Discrete SE) follow closely but remain less effective than SRE with UQ, while probability-based ($p$(True)) and embedding regression baselines lag significantly behind.
2. The advantage of SRE with UQ is particularly pronounced in challenging datasets such as NQ and SQuAD, where its RAC curves show clear separation from both semantic and non-semantic baselines, highlighting robustness under high variability.
3. Moving from 16-bit to 4-bit precision induces only minor changes in the RAC slopes of SRE with UQ, whereas competing methods (e.g., SE and NE) degrade more noticeably. This underscores the resilience of SRE with UQ to quantization.
4. Model size plays a role, with smaller models (e.g., LLaMA-3.2-1B) exhibiting flatter RAC curves than larger counterparts (e.g., LLaMA-2-13B). Nonetheless, SRE with UQ consistently preserves its relative advantage across all scales, confirming its effectiveness even under reduced representational capacity.
5. We also observe a minor performance drop for the NQ dataset when using the LLaMA 3.1B model, consistent with the results in C.1.1, which warrants further investigation in future work.

### C.2.2 RAC PERFORMANCE: PHRASE LENGTH ON INSTRUCT MODELS

Figs. 18, 19, and 20 present RAC curves at 16-bit, 8-bit, and 4-bit precision, respectively, for six hallucination quantification methods across multiple LLMs and four datasets under phrase-length generation on instruct fine tuned models. Phrase-level completions are particularly relevant for instruction-following use cases, where responses tend to be short and context-sensitive. While prior SOTA hallucination detection methods have largely overlooked this setting, understanding rejection–accuracy behavior under compressed instruction-tuned models is crucial for safe deployment in real-world applications.

This analysis reveals how quantization interacts with model adaptation to instruction tuning, and whether semantic-based uncertainty measures such as SRE with UQ retain their reliability when generations are concise and more sensitive to uncertainty calibration. In total, we conducted **8 experiments** spanning all quantization levels, providing a comprehensive basis for phrase length analysis, drawing parallel with its AUROC counterpart evaluation (Appendix C.1.1). Several notable patterns emerge upon analysis:

1. The proposed SRE with UQ consistently yields the steepest RAC slopes across datasets, demonstrating stronger rejection behavior than both probability-based ($p$(True)) and embedding regression baselines.
2. Instruct models reveal that semantic criteria—especially SRE with UQ—retain substantially higher accuracy at rejection rates above 90%, with the advantage most pronounced on challenging datasets such as TriviaQA and NQ.

Figure 15: **summarizing 12 experimental** scenarios, RAC scores for confabulation detection across three LLMs (LLaMA 3.2 1B, LLaMA 2 7B, LLaMA 2 13B) at 16 bit precision and four datasets (TriviaQA, SQuAD, NQ, SVAMP). The performance of the proposed SRE with UQ is in par or even higher than SOTA methods.

3. Across quantization levels, SRE with UQ remains stable from 16-bit to 4-bit precision, whereas baselines degrade more noticeably, underscoring its resilience to model compression.

4. Larger instruct model (e.g., LLaMA-2-13B-chat) amplify the gains of SRE with UQ, while smaller model (e.g., LLaMA-2-7B-chat) show flatter RAC curves; nonetheless, SRE with UQ consistently outperforms alternatives across scales.

### C.2.3 RAC PERFORMANCE: PHRASE LENGTH ON NON INSTRUCT MODELS

Phrase-level RAC plots for non-instruct LLMs (Figs. 21, 22, and 23) provide further evidence that semantic-based hallucination detection, and in particular our proposed SRE with UQ, maintains strong robustness under quantization and across diverse datasets. Unlike token-probability and embedding-based baselines, which exhibit flatter or unstable RAC slopes, SRE-UQ consistently yields steeper rejection–accuracy improvements, highlighting its ability to reliably prioritize correct outputs even when compression reduces representational capacity. These findings extend the AUROC trends to phrase-level rejection settings, demonstrating that SRE-UQ is not only effective in distinguishing confabulations but also excels in dynamically filtering them under realistic deployment constraints. Importantly, the proposed SRE with UQ demonstrates consistently stronger rejection–accuracy behavior, highlighting its suitability for lightweight non-instruct models where calibration is more challenging. Several notable patterns emerge upon analysis:

1. SRE-UQ leads across precisions, across 16-bit, 8-bit, and 4-bit quantization, SRE-UQ produces the steepest RAC curves, confirming that it most effectively raises accuracy as rejection increases. This advantage is especially evident in TriviaQA and NQ, where probability-based baselines stagnate or even flatten.

2. While baselines such as Discrete Semantic Entropy and Naïve Entropy show visible degradation under 4-bit quantization, SRE-UQ's RAC slopes remain sharp and consistent, underscoring resilience to aggressive model compression.

3. Gains are most pronounced in datasets with higher linguistic variability (e.g., NQ and SVAMP), where SRE-UQ sharply separates from weaker baselines. By contrast, probability-based $p(\text{True})$ often collapses under phrase-level rejection, failing to provide meaningful filtering.

4. Smaller models (e.g., LLaMA-3.2-1B) exhibit generally flatter curves due to limited representational power, but even in these challenging cases, SRE-UQ maintains relative gains, outperforming all other methods consistently.

Figure 16: **summarizing 20 experimental** scenarios, RAC scores for confabulation detection across five LLMs (LLaMA 2 13B, LLaMA 2 7B, LLaMA 3.2 1B, Mistral-7B, Mistral-7B-chat) at 8 bit precision and four datasets (TriviaQA, SQuAD, NQ, SVAMP). The performance of the proposed SRE with UQ is in par or even higher than SOTA methods.

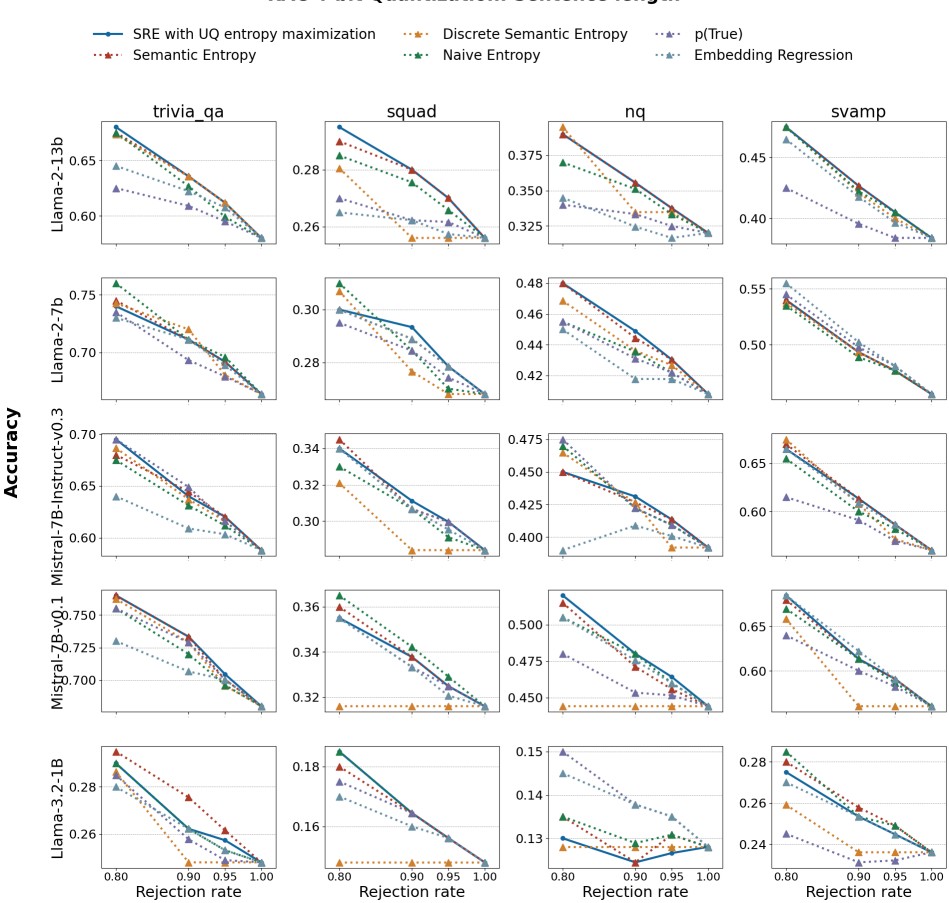

Figure 17: **summarizing 20 experimental** scenarios, RAC scores for confabulation detection across five LLMs (LLaMA 2 13B, LLaMA 2 7B, LLaMA 3.2 1B, Mistral-7B, Mistral-7B-chat) at 8 bit precision and four datasets (TriviaQA, SQuAD, NQ, SVAMP). The performance of the proposed SRE with UQ is in par or even higher than SOTA methods.

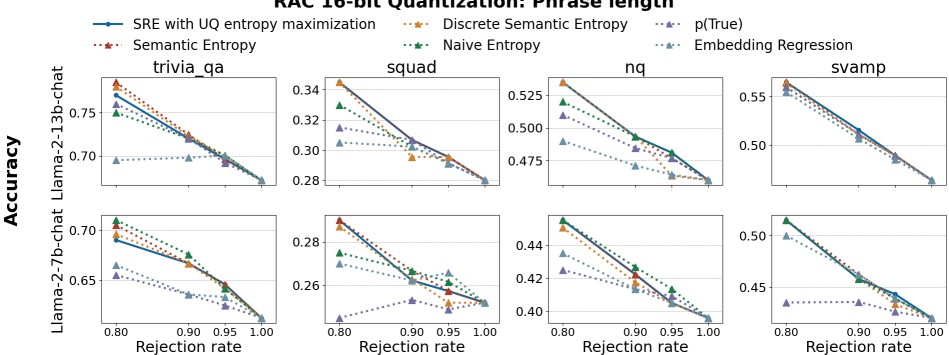

Figure 18: **summarizing 8 experimental** scenarios, RAC scores for confabulation detection across two LLMs (LLaMA 2 13B-chat, LLaMA 2 7B-chat) at 16 bit precision and four datasets (TriviaQA, SQuAD, NQ, SVAMP). The performance of the proposed SRE UQ is in par or even higher than SOTA methods.

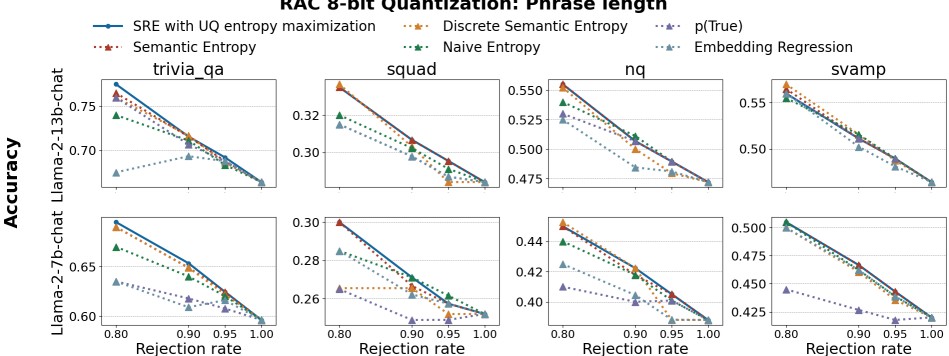

Figure 19: **summarizing 8 experimental** scenarios, RAC scores for confabulation detection across two LLMs (LLaMA 2 13B-chat, LLaMA 2 7B-chat) at 8 bit precision and four datasets (TriviaQA, SQuAD, NQ, SVAMP). The performance of the proposed SRE UQ is in par or even higher than SOTA methods.

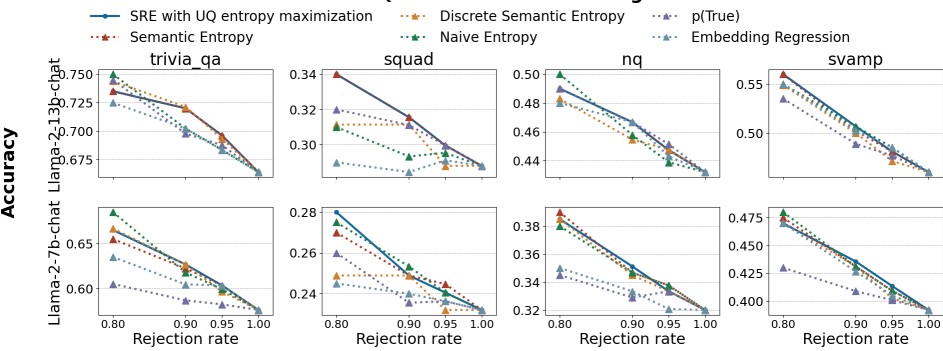

Figure 20: **summarizing 8 experimental** scenarios, RAC scores for confabulation detection across two LLMs (LLaMA 2 13B-chat, LLaMA 2 7B-chat) at 4 bit precision and four datasets (TriviaQA, SQuAD, NQ, SVAMP). The performance of the proposed SRE UQ is in par or even higher than SOTA methods.

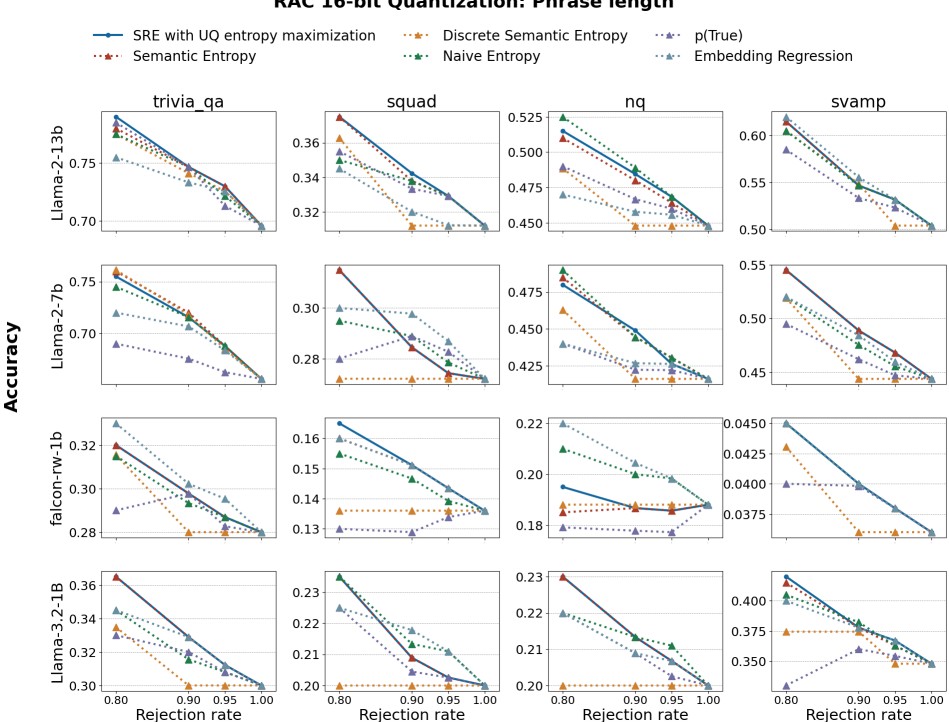

Figure 21: RAC scores for confabulation detection across four LLMs (Mistral-7B, Falcon-1B, LLaMA 3.2B, LLaMA 2 7B 4-bit) and four datasets (TriviaQA, SQuAD, NQ, SVAMP). The performance of the proposed semantic Rényi entropy is in par or even higher than SOTA methods.

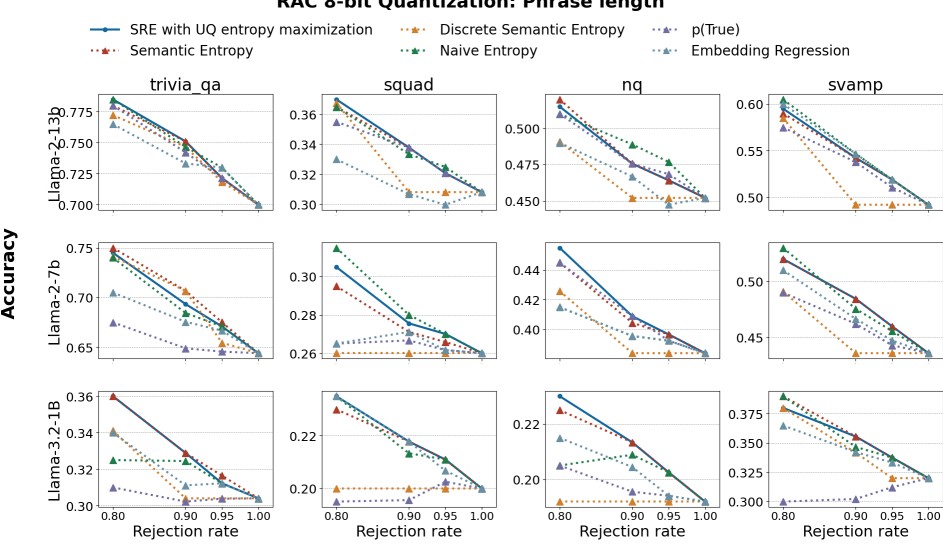

Figure 22: RAC scores for confabulation detection across four LLMs (Mistral-7B, Falcon-1B, LLaMA 3.2B, LLaMA 2 7B 4-bit) and four datasets (TriviaQA, SQuAD, NQ, SVAMP). The performance of the proposed semantic Rényi entropy is in par or even higher than SOTA methods.

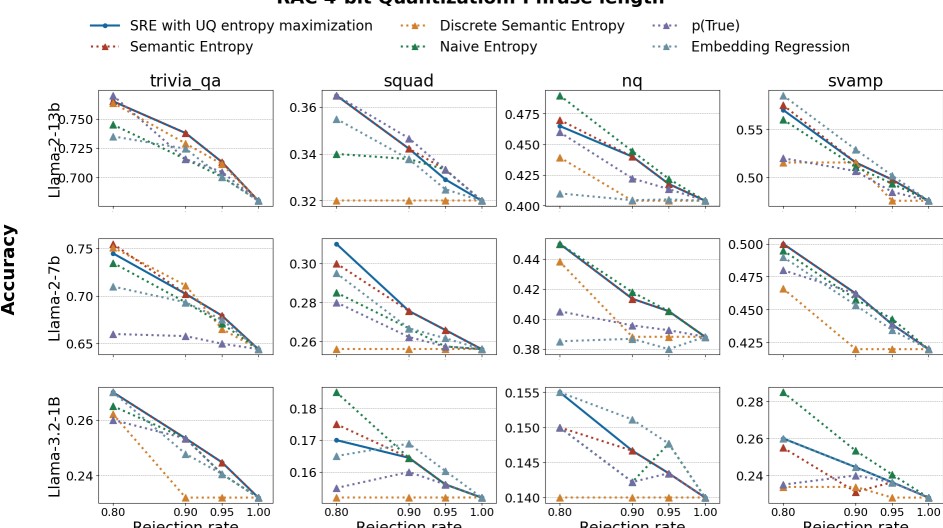

Figure 23: RAC scores for confabulation detection across four LLMs (Mistral-7B, Falcon-1B, LLaMA 3.2B, LLaMA 2 7B 4-bit) and four datasets (TriviaQA, SQuAD, NQ, SVAMP). The performance of the proposed semantic Rényi entropy is in par or even higher than SOTA methods.

## C.3 HYPERPARAMETER SELECTION

In this section, we examine the effect of hyperparameter tuning for $\lambda$ in the entropy maximization framework introduced in equation 7. Recall that this formulation adjusts the output TS probabilities by jointly maximizing semantic entropy and penalizing deviations from the original model distribution, weighted by the model's uncertainty estimates. A critical innovation of our approach lies in integrating UQ directly into the token adjustment mechanism, thereby enhancing the model's sensitivity to confabulations beyond conventional semantic entropy analysis.

To assess the impact of $\lambda$, we measure AUROC scores for hallucination detection across different values of the hyperparameter. In each case, the Rényi's semantic entropy is computed after probability adjustment, and the results are compared against the baseline Rényi's entropy without uncertainty correction (denoted by the red line). The $\lambda$ yielding the highest AUROC is in green. This comparison isolates the contribution of UQ-aware entropy maximization to the confabulation detection task.

As shown in Fig. 24, incorporating UQ-based probability adjustments yields substantial improvements over the baseline entropy approach across all models tested.

Several key observations are notable:

1. Moderate values of $\lambda$ systematically boost AUROC scores, confirming that a calibrated trade-off between entropy maximization and fidelity to the original model predictions is beneficial.

2. While the precise optimal $\lambda$ may vary slightly across models and datasets, the performance gains are robust across a wide range of $\lambda$ values, underscoring the general effectiveness of the proposed correction.

3. This experiment validates one of the most crucial contributions of our work: by introducing UQ into semantic entropy calculations, we can significantly sharpen the model's ability to discriminate confabulations, especially in the presence of noisy or low-confidence predictions.

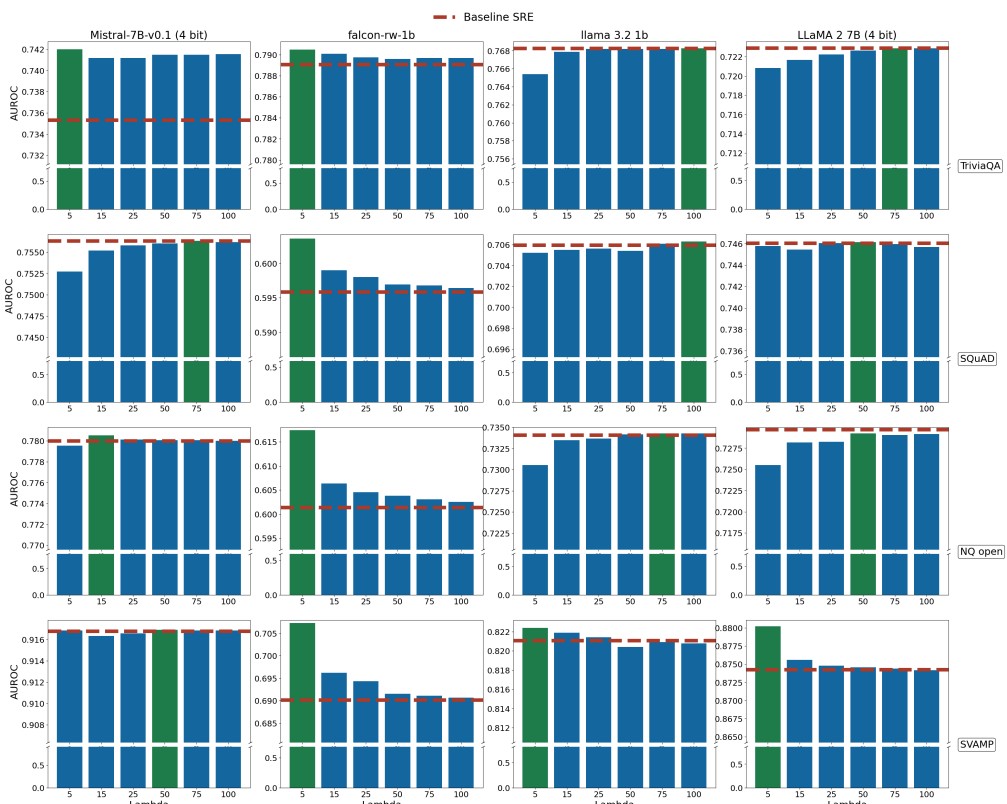

Figure 24: Effect of $\lambda$ hyperparameter selection on AUROC performance. The red line denotes the baseline SRE without UQ adjustment. Integrating uncertainty via equation 7 consistently enhances hallucination detection capability across all LLM models and datasets. The $\lambda$ yielding the highest AUROC is in green.

# D ADDITIONAL DETAILS ON ALGORITHMIC STRUCTURES AND COMPUTATIONAL COST

## D.1 COMPUTATIONAL OVERHEAD

The computational overhead introduced by the QTN-based UQ module is modest and remains well within real-time inference constraints. The method requires storing only $M = 8$ q-bit perturbation features in a $2^8 \times 2^8$ matrix, which is significantly smaller than the embedding caches required by clustering-based semantic entropy methods. Constructing the Hamiltonian associated with the sampled KME incurs a *one-time* pre-processing cost of approximately 45–60 s on CPU, though this can be substantially reduced through GPU-accelerated linear algebra routines. At inference time, the per-query overhead is minimal: evaluating the perturbation-based uncertainty features adds merely 6–10 ms on an NVIDIA A6000 GPU, which is negligible compared to the intrinsic latency of LLM decoding. Overall, the QTN perturbation step introduces only a lightweight computational cost while enabling richer uncertainty estimates.

## D.2 ALGORITHMIC STRUCTURE AND COMPLEXITY

In this subsection, we present pseudocode illustrating key components of our approach, complementing the pipeline diagram provided in Fig. 1.

This includes pseudocode for (i) Hamiltonian construction from the sampled KME, (ii) extraction of first-order perturbation features, and (iii) the entropy-maximization procedure from Sec. 2.3.

---

**Algorithm 1** Construction of Empirical KME $\widehat{\psi}(x)$

---

1: **Input:** Samples $\{x_i\}_{i=1}^N$ from random variable $X$; kernel bandwidth $\sigma$; sampling grid $\{x_j\}_{j=0}^{M-1}$
2: **Output:** Sampled KME vector $\widehat{\psi} = \{\widehat{\psi}_j\}_{j=0}^{M-1}$
3: **for** $j = 0$ to $M - 1$ **do**
4:     Compute Parzen PDF estimate:

$$\widehat{p}(x_j) = \frac{1}{N} \sum_{i=1}^N \kappa_\sigma(x_j; x_i)$$

    where $\kappa_\sigma(x_j; x_i)$ is the Gaussian kernel
5:     Set KME value $\widehat{\psi}_j \leftarrow \widehat{p}(x_j)$
6: **end for**
7: **return** $\widehat{\psi}$

---

**Algorithm 2** QTN Hamiltonian Construction via Quantum Correlation Matrix

---

1: **Input:** Sampled KME vector $\widehat{\psi}$ of length $M = 2^L$; operator basis $\widehat{\mathcal{H}} = \{\widehat{H}_k\}_{k=0}^{T-1}$
2: **Output:** Local Hamiltonian $\widehat{H}$
3: Construct the QCM matrix using the inner products, where the $(k, \ell)$ entry is given by :

$$\mathbf{M}_{\widehat{\psi}}(k, \ell) = \langle \widehat{H}_k \widehat{\psi}, \widehat{H}_\ell \widehat{\psi} \rangle$$

4: Compute null space of QCM:

$$\mathbf{w} = \{w_k\}_{k=0}^{T-1} \in \mathrm{Null}(\mathbf{M}_{\widehat{\psi}})$$

5: Form Hamiltonian as linear combination:

$$\widehat{H} = \sum_{k=0}^{T-1} w_k \widehat{H}_k$$

6: **return** $\widehat{H}$

---

---

**Algorithm 3** First-Order Perturbation Corrections and TS UQ

---

1: **Input:** Base Hamiltonian

$$\widehat{\boldsymbol{H}} = \sum_{k=0}^{T-1} w_k \, \widehat{\boldsymbol{H}}_k,$$

where $\{w_k\}$ are the learned weights multiplying the Hermitian operator basis; perturbation vector $\Delta \mathbf{w}$ specifying perturbation of these weights.

2: **Output:** First-order corrected eigen-modes $\{\psi_m^{(1)}\}$ and the resulting UQ feature vector.

3: Construct the perturbed Hamiltonian:

$$\Delta \boldsymbol{H} = \sum_{k=0}^{T-1} \Delta w_k \, \widehat{\boldsymbol{H}}_k.$$

This corresponds to perturbing the coefficients (weights) associated with the Pauli-like operators forming the QTN Hamiltonian.

4: Compute eigen-decomposition of the unperturbed Hamiltonian:

$$\widehat{\boldsymbol{H}} \, \psi_m = E_m \, \psi_m, \qquad m = 0, \dots, M-1.$$

5: **for** $m = 0$ to $M - 1$ **do**

6:     Compute first-order energy correction:

$$E_m^{(1)} = \langle \psi_m, \, \Delta \boldsymbol{H} \, \psi_m \rangle.$$

7:     Compute first-order eigenvector correction:

$$\psi_m^{(1)} = \sum_{n \neq m} \frac{\langle \psi_n, \, \Delta \boldsymbol{H} \, \psi_m \rangle}{E_m - E_n} \, \psi_n.$$

8: **end for**

9: We utilize Eq. 5 and $\{\psi_m^{(1)}\}$ to compute UQ feature vectors in the KME amplitude domain, yielding the local UQ scores that appear in Eq. 6.

10: **return** $\{\psi_m^{(1)}\}$ & $\mathrm{UQ}(p_s^{(r)})$.

---

---

**Algorithm 4** Calibrated Entropy-Maximization Update for TS Probabilities

---

1: **Input:**
   - Original TS probability $p_s^{(r)}$
   - Uncertainty score $\mathrm{UQ}(p_s^{(r)})$ computed via Eq. 6
   - Regularization parameter $\lambda$

2: **Output:** Adjusted probability $p_s^{(r)^*}$

3: Define objective:

$$\mathcal{L}(\widehat{p}) = -\log\big(\widehat{p}^2 + (1-\widehat{p})^2\big) - \frac{\lambda}{\mathrm{UQ}(p_s^{(r)})} \cdot \mathrm{KL}\Big(\widehat{p} \,\|\, p_s^{(r)}\Big)$$

where $\mathrm{KL}(\widehat{p}\|p) = \widehat{p} \log \frac{\widehat{p}}{p} + (1-\widehat{p}) \log \frac{1-\widehat{p}}{1-p}$.

4: Initialize: $\widehat{p}^{(0)} \leftarrow p_s^{(r)}$

5: **for** $t = 1$ to $T_{\max}$ **do**

6:     Compute gradient $\nabla \mathcal{L}(\widehat{p}^{(t-1)})$

7:     Update via projected gradient ascent

8:     Stop early if $|\widehat{p}^{(t)} - \widehat{p}^{(t-1)}| < \delta$

9: **end for**

10: Set $p_s^{(r)^*} \leftarrow \widehat{p}^{(T)}$

11: **return** $p_s^{(r)^*}$

---

