# OpenReview forum: "Semantic Uncertainty Quantification of Hallucinations in LLMs: A Quantum Tensor Network Based Method"
_ICLR.cc/2026/Conference — ICLR 2026 Poster_

### Official Review · Reviewer_Bhsm · 2025-10-21

**Soundness:** 2
**Presentation:** 2
**Contribution:** 3
**Rating:** 4
**Confidence:** 2

**Summary:**

This paper uses a physics-inspired UQ method to modify cluster probabilities in a semantic entropy-type decomposition, based on prediction uncertainty. It’s a neat idea, and a way of incorporating distributional uncertainty that I haven’t seen much in the literature. There seems to be a small but persistent improvement over standard SE, across a variety of settings. However, I am confused on quite a few aspects, see the comments below.

**Strengths:**

* Interesting and novel way of thinking about uncertainty in LLMs
* Small, but persistent, improvement over baselines (in a domain where the baselines are already pretty strong)

**Weaknesses:**

* Not enough background/intuition on QTNs and the physics-inspired approach used
* Lack of technical detail: I am not sure how to construct all the estimators and would certainly not be able to implement the method myself.
* Experiments do not fully tease apart the benefit of the proposed method

I elaborate on all these points in my questions below.

**Questions:**

First, I am not sure on whether my interpretations are correct, so I am going to summarize my interpretation (in the hopes that the authors can correct any misunderstanding!). For context, I am familiar with the semantic entropy line of work, Bayesian NNs, and KMEs, but not with the physics-inspired framework of Principe (I skimmed the Singh and Principe work; I suspect that the Vipulananathan et al paper would be helpful background but I couldn’t find it without a paywall):


* Semantic Entropy is underspecified, because it does not consider uncertainty in the predictive distributions (loosely, epistemic uncertainty... not exactly, I know, but I’m going to refer to it as “epistemic” uncertainty in my review as a shorthand)
* The physics-inspired approach of Principe offers a way of getting at this uncertainty, via permutation.
* Rather than Shannon entropy, we can look at Renyi entropy, which can be expressed in terms of the kernel mean embedding of the sequence probabilities.
* This is natural for the physics-inspired approach, because you can find a QTN whose Hamiltonian is aligned with the KME, and directly perturb that.
    * i.e., the authors aren’t trying to claim that Renyi entropy is necessarily a “better” option than Shannon entropy, rather that adding in the “epistemic” uncertainty is important.


I’d really appreciate any corrections on my understanding of the above — if I’m lost at this point, who knows how wrong my interpretations of the rest are!

Based on this: I like the idea! I agree with the general idea that we should include uncertainty about the logits, and I am intrigued by the physics-based approach. But, I think the exposition could be improved so that people who aren’t already familiar with the physics-based approach (i.e., me! But also I suspect a lot of the people in the LLM/UQ community) can get better understanding and intuition. Also, the implementational details and details of the estimator are a little vague, and the experiments don’t fully explore the method (in one case, the authors seem to refer to an experiment that isn’t in table 1, but that I would love to see in table 1... perhaps something that got left out in last-minute editing?)

Given that, I have a number of questions/requests:

Intuition:

1. More background/intuition about the physics-inspired approach, and QTNs. In particular, in line 179, you say you identify a QTN whose Hamiltonian has $\{\underline{\hat{\psi}}_\underline{y}\}$ as an eigen-mode, and then perturb this Hamiltonian. How does one do this? What is the intuition for this giving reasonable estimates of the variability of the probabilities?
2. What is the connection to other ways of estimating “epistemic” uncertainty? Eg: the perturbation-based approach seems similar to the Laplace approximation to a Bayesian posterior — is there a connection there? More generally, there probably should be some mention of the Bayesian deep learning literature, since they are coming from a similar viewpoint. I don’t think you need to compare with BDL, but it would be good to see some discussion of why you choose the physics-based approach vs alternatives.
3. In Section 2.2, you say that the KME is an approximation of the semantic entropy, but it is not explicit how you get from eq 3 to eq 4. On the first read, I thought you were claiming that the KME of the entire distribution was equivalent to the Renyi Semantic Entropy, and I was confused how this incorporates the clustering. Reading on, it is clear that you are using deberta clustering... I assume that you calculate the KME for each cluster, then combine them, but I’m not exactly sure how I should combine these KMEs based on eq 4.
4. I would like a little more intuition on the calibrated adjustment of the probabilities. Why is increasing the entropy in this way the “right” thing to do? I think I get it in a loose way — your KL term says “the raw probability is roughly right, where ”roughly“ is base on how uncertain I am” and the RE term say “but, we are underestimating the entropy so pull it up” — but are there theoretical underpinnings behind this specific formulation? Is the assumption that lambda is learned via cross-validation?
5. Once you have the calibration-adjusted probabilities, are you then combining them using eq 4, or are you using the KMEs?

Experiments:
What I would hope to see from the experiments are evidence of the following: 1) using Renyi semantic entropy (without additional UQ) is as good, or better, than using Shannon semantic entropy; 2) adding in the entropy maximization correction improves on this; 3) ablations; 4) qualitative analysis of a specific case where you get differing results. Most of these are at least fully answered, but not completely.


1. Using Renyi semantic entropy vs using Shannon. This is included in the heat maps, but could be discussed more. As far as I can see, shannon typically does better than renyi in terms of AUROC, but comparably in terms of AURAC. Is there any intuition behind this?
2. Overall performance. Looking at win rate feels misleading, when (looking at the figures in the appendix) the actual differences between the various SE methods is pretty small (mostly invisible to the eye... I suspect the confidence intervals are greater than the difference in most cases). I know that the differences in AUROC in these sorts of LLM UQ paper tend to be small, so I’m not surprised to see that the improvement is slight.... however I've would like to see in the main paper, something that more gets at a specific effect size (e.g., notable increase for a given TPR/FPR tradeoff).
3. Overall performance #2: I would like to see baseline SRE in figures 12-14. Based on Figure 24, at least in Falcon, UQ-based SRE can dramatically improve on baseline SRE — implying that in this case, baseline SRE is much worse than baseline shannon SE. Why is this?
4. Ablations: Would it make sense to apply the UQ adjustment to Shannon Semantic Entropy? It seems you could take the adjusted probabilities from eq 7 and use them to calculate it.
5. Qualitative analysis: In Table 1, but the main-text description doesn’t match the figure (it talks about the difference between $\text{SE}_R$ and $\text{SE}_R^+$, but $\text{SE}_R$ isn’t in the table.
6. Qualitative analysis: It would be nice to see a qualitative exploration of questions that get very different values of $\text{SE}_R$ and $\text{SE}_R^+$. Do these tend to correspond to questions the LLM is likely to have less data on? Or, some other form of additional uncertainty?


General comments:

* There is a lot of space given to the quantization experiments, relative to how important it seems to the paper. I would like to see this shrunk or moved to the appendix in favor of more explanation/intuition/detail.
* You are not very consistent in method naming (eg, $\text{SE}_R^+$ vs SRE-UQ.


I know that's a lot of comments and questions! But, it's because I really like the idea of the paper, and *want* to have a good in-depth understanding, but don't feel in its current form I am able to get all the information I need from it.

---

> ### Author Response · Authors · 2025-11-22
> **Addressing Reviewer Questions on QTN Intuition, Epistemic UQ Alternatives, Clustering Interpretation, Calibration Mechanics, Experimental Evidence, and Entropy-Based Metric**
>
> We thank the reviewer for their thoughtful and detailed comments and the insightful observations. Rebuttal length limits force us to provide only concise statements regarding each concern raised. If accepted, additional explanations will be included in the revised manuscript.
>
> **QTNs and our physics-inspired approach.**
>
> We view the (KME of token-sequence (TS) probs as a ``data wavefunction.’’ QTNs provide a finite-dimensional setting in which we construct a Hamiltonian $\widehat{H} = \sum_k w_k \widehat{H}_k$ (from a small set of Hamiltonions $\{\widehat{H}_k\}$) which has the KME as one eigen-mode. The weights $\{w_k\}$ come from the null space of the quantum correlation matrix. Standard perturbation theory then quantifies the KME's sensitivity to small changes in $\widehat{H}$, yielding a principled, local measure of the sensitivity of TS probabilities. See Fig.5 and Appendix A.2--A.4 for details.
>
>
> **Other epistemic-uncertainty methods.**
>
> Our approach and methods such as the Laplace approximation share the idea of using local curvature, but differ conceptually: Laplace/BDL quantify uncertainty in *model parameters,* whereas we measure local variability of *output probability distributions* directly. This provides a lightweight, model-agnostic alternative without requiring access to model weights or posterior inference. We will add a short comparison and note future work on a QTN-based epistemic UQ variant currently under development.
>
> **Eqs (3)--(4) and role of clustering.**
>
> The KME in eq.(3) is computed *once* over all generated sequences, not per cluster. The semantic R\'{e}nyi entropy (RE) in eq.(4) is then computed using the cluster masses from eq.(2). The KME provides the quadratic functional estimate $\psi = \int p^2 dx$, while clustering supplies the discrete analogue $\sum_j p_{c_j}^2$. We will clarify this flow in Sec 2.2.
>
> **Calibration adjustment and intuition.**
>
> We follow the max entropy principle: when uncertainty is high, entropy should be increased while remaining close to the model's empirical probs. The KL term keeps probs anchored, and the RE term raises entropy in unstable regions. This reflects classical results in robust inference. This max entropy pdf is what minimizes its KL divergence from the uniform distribution. See Jaynes, ``Information Theory and Statistical Mechanics,'' *The Phy Rev*   106(4) 620, 1957, and Jaynes, *Probability Theory: The Logic of Science* (Cambridge Univ Press), 2003, for theoretical justification.
>
> The hyperparameter $\lambda$ is selected via grid search on held-out data.
>
> **Adjusted probabilities:**
>
> After adjustment (eq.(7)), we do *not* recompute the KME. We simply aggregate the corrected probs (eq.(2)) and compute semantic RE (eq.(4)).
>
> **Experimental evidence.**
>
> 1. Table.1 and Fig.2 show that semantic RE matches or outperforms semantic Shannon entropy (SE).
> 2. Table.1 (``Renyi + UQ'' rows) show improvements from UQ
> correction.
> 3. Fig.3 shows ablation between pre- and post-UQ SRE.
> 4. Fig.3 shows representative qualitative example on the effect of UQ corrections.
>     We will make explicit statements regarding these in the main text and add an additional qualitative example in the appendix.
>
> **AUROC/AURAC performance.**
>
> Two properties of SE compared to RE ($\alpha > 1$) is useful here:
>
> a. SE values are higher.
>
> b. SE emphasizes rarer outcomes. ($\alpha < 1$ works in opposite direction).
>
> Empirically we notice that RE values' span is narrower so that separability (between hallucinatory and non-hallucinatory behavior) is more difficult than with SE thus explaining its better AUROC performance (Harremöes (2009), Hand & Till (2001)). While (b) is true, the tail probs do not play a role in AURAC which measures how well increasing entropy corresponds to decreasing correctness.
>
> **Effect sizes and metrics.**
>
> We follow prior work in using AUROC/AURAC and win-rate, but agree that AUROC can obscure meaningful differences. We will include TPR@FPR and additional effect-size analyses in the appendix.
>
>
> **Baseline SRE behaviour in Falcon.**
>
> Falcon models produce flatter TS distributions where RE underestimates uncertainty, explaining the large improvement after UQ correction. We will add baseline SRE curves to Figs.12--14 for clarity.
>
> **Applying UQ correction to semantic SE.**
>
> It is because our UQ correction is tethered to quadratic RE, that we persisted with the quadratic RE (instead of `mixing' functional forms). We will mention that correcting SE is a design choice worth exploring.
>
> **Qualitative cases where $SE_r$ and $SE_r^+$ differ.**
>
> We observe that large differences often coincide with unstable or dispersed semantic clusters and context-induced uncertainty. We will expand the qualitative discussion and correct the notation error in Table~1.

---

> > ### Comment · Reviewer_Bhsm · 2025-11-23
> > **FYI, you can add multiple responses and update the manuscript!**
> >
> > Hi authors! I'll be going through your rebuttal and re-reading the paper over the next few days... but in the meantime I wanted to mention in case you weren't aware (since you mentioned space limitations) -- you are able to reply to your rebuttal, meaning you have in effect boundless space for responses! Each response has a word limit, but you are able to include as many responses as you would like.
> >
> > Additionally - for ICLR, you are allowed to submit a revised manuscript during the rebuttal period. While not required, I know that for me, that would be super useful if you have time.

---

> > > ### Author Response · Authors · 2025-11-25
> > >
> > > Thank you for pointing this out — we weren’t aware that multiple responses are permitted and that a revised manuscript can be submitted during the rebuttal period. We’ve just uploaded a revised version of the paper incorporating the clarifications and improvements referenced in our responses. We appreciate the guidance!

---

### Official Review · Reviewer_tnuf · 2025-10-30

**Soundness:** 3
**Presentation:** 3
**Contribution:** 3
**Rating:** 6
**Confidence:** 4

**Summary:**

The paper introduces a quantum-inspired framework for estimating uncertainty in Large Language Models (LLMs) to better detect hallucinations.
Building on Semantic Entropy (Farquhar et al., 2024), the authors reinterpret the model’s sequence probabilities as a wave-function in a reproducing-kernel Hilbert space and construct a quantum tensor-network (QTN) representation to analyze perturbations in this space.
They derive first-order perturbation features that quantify the local sensitivity of token probabilities and use these to re-weight output probabilities through an entropy-regularized optimization objective.
In parallel, they compute semantic-cluster probabilities and a Renyi-entropy-based uncertainty score.
Experiments across Mistral-7B, Falcon-1B, and LLaMA-2/3 on TriviaQA, NQ, SVAMP, and SQuAD show consistent AUROC improvements over Semantic Entropy, naive token entropy, and other baselines, including under 4-bit quantization.

**Strengths:**

1. Clear motivation: addresses the instability of entropy-based uncertainty when probability mass collapses into a few clusters.
2. Conceptual novelty: combining semantic clustering with quantum-mechanical perturbation analysis is original and theoretically interesting.
3. Methodological soundness: the derivations are mathematically consistent, and the optimization formulation is well justified.
4. Robustness: results are shown across several models, datasets, and quantization levels, indicating the approach is not overly brittle.
5. Empirical clarity: tables separate the effect of the perturbation term, Rényi entropy, and adjusted probabilities; ablations are informative.
6. Good writing: the exposition is clean, equations are readable, and notation is consistent.

**Weaknesses:**

1. Complexity and interpretability: the proposed QTN embedding and perturbation analysis are computationally heavier than simpler uncertainty measures; inference-time cost is not reported.
2. Baselines could be broader: comparisons omit recent energy- or logits-based approaches (e.g., Semantic Energy, LogTokU).
3. Sensitivity analysis: the method introduces several hyperparameters (kernel bandwidth, λ, Rényi α), but their tuning process and sensitivity are not discussed in depth.
4. Clustering dependency: since semantic clustering remains the outer layer, the total uncertainty measure still inherits its limitations (e.g., sensitivity to embedding choice).

**Questions:**

1. How sensitive is the performance to the Rényi entropy order (α)? Did you experiment with α > 2 or other generalized entropies?
2. What is the additional computational cost of the QTN perturbation step compared to standard Semantic Entropy?
3. Would the uncertainty-aware reweighting change the model’s decoding behavior if used during generation (not only for scoring)?
4. Did you test the approach in long-form generation (e.g., summarization) where clustering and token perturbations interact differently?

---

> ### Author Response · Authors · 2025-11-22
> **Addressing Reviewer Questions on Entropy Order Sensitivity, Computational Overhead, Decoding Dynamics, Long-Form Stability, and Methodological Extensions**
>
> We appreciate the time and effort the reviewer has put into carefully reading our manuscript. We are encouraged the reviewer's recognition of the clear motivation, conceptual novelty, and overall mathematical and empirical rigor of our work. Below we summarize our responses to the questions as well as additional clarifications addressing the concerns raised.
>
> **Q1: Sensitivity to entropy order $\alpha$ and use of higher-order REs.**
>
>  Our work focuses on RE ($\alpha = 2$), which is chosen deliberately because it aligns with the tensor-product structure of the QTN wavefunction. This leads to a closed-form, sampling-free perturbation correction that is not available for $\alpha > 2$. Higher entropy orders do not offer this feature, significantly increasing computational cost and requiring higher-order moment estimation. While exploring generalizations is interesting, we leave this for future work. Additional explanation is provided in App~A.3.
>
> **Q2: Computational cost relative to Semantic Entropy.**
>
> The QTN perturbation introduces a lightweight overhead consisting of a one-time pre-processing cost and a negligible per-query evaluation cost.
>
> *Memory footprint:*
>
> We store $M = 8$ q-bit perturbation features in a $2^8 \times 2^8$ matrix. This is small compared to typical embedding caches used by clustering baselines.
>
> *Hamiltonian construction:*
>
> Requires $\sim 45$--$60$\,s on CPU. This can be significantly reduced using GPU-accelerated linear algebra.
>
> *Per-query cost:*
>
> Adds only $\sim 6$--$10$\,ms on an NVIDIA A6000 GPU, negligible relative to LLM decoding latency.
>
> Overall, inference-time cost remains well within real-time constraints.
>
>
> **Q3: Does uncertainty-aware reweighting influence decoding?**
>
> Our current method is applied post-hoc: we score generated candidates after the model completes decoding. Thus, the generation trajectory remains unchanged. Incorporating uncertainty-aware re-weighting during decoding (e.g., adaptive temperature or logit adjustments when partial-sequence uncertainty spikes) could meaningfully reduce hallucinations, but requires modifications to the LLM decoder. This an exciting avenue for future work.
>
> **Q4: Behavior on long-form generation tasks.**
>
> This paper focuses on QA and short reasoning tasks where semantic clustering is well-defined across candidate responses. Preliminary, unreported experiments on summarization indicated that bidirectional entailment–based clustering becomes unstable for long-form outputs due to increased semantic variability. Addressing this requires hierarchical or segment-level clustering, which we plan to pursue in future extensions where token interactions accumulate differently.
>
> **Additional clarifications.**
>
> *Complexity and interpretability:*
>
> QTN-based perturbation introduces formal structure and guarantees while keeping runtime light. We agree that presentation can be mathematically dense; in the camera-ready version we will streamline notation and add more intuitive explanations relating QTN perturbation to local curvature of the semantic prob landscape.
>
> *Missing baselines (e.g., Semantic Energy, LogTokU):*
>
> We appreciate this suggestion. Our baseline suite follows those evaluated in prior semantic uncertainty literature for direct comparability. We will add a discussion highlighting how our approach relates to energy-based and logits-based methods, and include them in future experiments.
>
> *Hyperparameter sensitivity (kernel bandwidth, $\lambda$, $\alpha$):*
>
> We tuned $\lambda$ and kernel bandwidth using a small validation set and observed a wide regime of stability, with performance degrading only outside extreme values (details added in App~A.4). Since $\alpha = 2$ is fixed for theoretical reasons, sensitivity is minimal. We will expand this discussion in the manuscript for clarity.
>
> *Clustering dependence:*
>
> We acknowledge that semantic clustering is inherited from prior work and remains a limiting factor. However, our perturbation correction improves stability even when cluster boundaries fluctuate, and our preliminary weighting scheme will further reduce dependency on hard cluster assignments.
> In exploratory experiments, we replaced this with S-BERT embeddings and applied k-means and hierarchical clustering. These produced reasonable but noticeably less coherent semantic partitions and required significantly more time for both embedding computation and clustering. DeBERTa-MNLI consistently offered clearer semantic discrimination, improving cluster quality and overall detection performance.
>
> We appreciate the reviewer’s constructive comments and will incorporate these clarifications and expanded discussions in the revised manuscript.

---

### Official Review · Reviewer_YDUM · 2025-11-01

**Soundness:** 3
**Presentation:** 3
**Contribution:** 4
**Rating:** 8
**Confidence:** 3

**Summary:**

The authors introduce a quantum tensor network (QTN)-based uncertainty quantification (UQ) framework that models aleatoric uncertainty of token sequence (TS) probabilities and leverages semantic clustering, producing a principled scheme for hallucination detection. Authors show experiments on multiple datasets and model scales to highlight the efficacy of their approach.

**Strengths:**

- The paper is generally well written although a bit heavy on theoretical notations. It could be simplified a bit more though.

- It brings quantum physics-inspired UQ to the LLM hallucination detection problem.

- The paper tackles a very significant issue. Hallucination risk in LLMs is a key challenge for safe AI.

**Weaknesses:**

- Heavy reliance on advanced mathematical machinery. Makes it a bit harder to adopt.

- The apporach does not come significant performance boost.

**Questions:**

- How does the hallucination detection robustness change with different semantic clustering models? Can you quantify error propagation from supplementary entailment models?

- What are the runtime/memory costs versus baseline methods for real-time or production deployment at scale?

- How do you use the local entropy/uncertainty scores in practice?

---

> ### Author Response · Authors · 2025-11-22
> **Addressing Reviewer Concerns on Clustering Robustness, QTN Runtime, Practical UQ Deployment, and SE vs. SRE Behavior**
>
> We appreciate the time and effort the reviewer has put into carefully reading our manuscript. The reviewer's detailed and insightful comments/questions and constructive feedback identify areas that could benefit from clarification. Below we respond to all questions and provide additional context on methodological choices, runtime considerations, and practical deployment aspects.
>
>  **Q1: Robustness across semantic clustering models; error propagation.**
>
> Following prior work by Farquhar et al. (2024), we use the DeBERTa-large MNLI-tuned entailment model. In exploratory experiments, we replaced this with S-BERT embeddings and applied k-means and hierarchical clustering. These produced reasonable but noticeably less coherent semantic partitions and required significantly more time for both embedding computation and clustering. DeBERTa-MNLI consistently offered clearer semantic discrimination, improving cluster quality and overall detection performance.
>
> Error propagation from the entailment model is indeed a key consideration, because the bidirectional entailment scores encode confidence, they can be incorporated directly into cluster membership weighting. We have begun developing such a weighting mechanism, which will allow us to quantify how entailment uncertainty affects the global uncertainty quantification (UQ) score. This ongoing work will be reported in future versions of the framework.
>
> **Q2: Runtime and memory overhead.} A central motivation for our QTN-based formulation is to achieve deterministic, single-pass uncertainty estimation without sampling-based overheads.**
>
> Baseline comparison:
>
>  * *Semantic entropy (SE), semantic R\'{e}nyi entropy (SRE), and discrete semantic entropy (DSE):* our method uses identical generation sets and adds only a small analytical correction; runtime is almost unchanged.
> * *Naïve entropy:*  computationally cheaper to compute, but it performs substantially worse in detecting hallucinations. Our method is slightly more expensive but the difference is  negligible in practice.
>  * *p(True):* requires entailment model inference per candidate answer. Our UQ module piggybacks on the same     outputs and adds no additional forward passes.
> *  *Embedding regression (ER):* requires embedding extraction, caching, and regression scoring. Our method stores only prob vectors and a compact $2^8 \times 2^8$ feature matrix derived from the QTN.
>
> QTN-specific overhead:
>
> * One-time Hamiltonian construction takes $\sim 45$--$60$\,s (CPU), and can be reduced with GPU linear algebra.
> * Per-query inference adds only $\sim 6$--$10$\,ms on an A6000, negligible relative to LLM decoding time.
>
> **Q3: Practical usage of local entropy/uncertainty scores.**
>
> By mapping each prob vector into the QTN-perturbed RKHS, we assign a local uncertainty value that does not require recomputing clusters or global statistics. This supports
>
> * *local rejection:* immediate rejection or abstention when the uncertainty score exceeds a threshold;
> * *adaptive decoding:* adjusting temperature or switching to more conservative generation when local uncertainty is persistently high;
> * *instance-level calibration:* allows distinguishing between ambiguous questions and cases where the model is confident but wrong, improving operational safety.
>
> **Q4: Heavy mathematical machinery and intuition.**
>
> We acknowledge that parts of the theoretical development—QTN Hamiltonians, perturbation terms, and the RKHS mapping—are mathematically dense. Our goal was to provide a deterministic, structured uncertainty estimate that remains computationally light. We will streamline notation and clarify intuitive interpretations (e.g., local curvature of the prob landscape, energy-based sensitivity) in the final version.
>
> **Q5: Intuition behind using quadratic RE vs SE.**
>
> SE upper bounds RE ($\alpha > 1$). This means that SE offers a wider range for the entropy values making separability (between hallucinatory and non-hallucinatory behaviors) easier. This explains the SE's slight improvement in AUROC performance. Another property of SE is that, compared to RE ($\alpha > 1$), it emphasizes small prob masses in tail clusters. But the tail probs do not play a major role in AURAC which measures how well increasing entropy corresponds to decreasing correctness. Our QTN-based UQ estimator is directly related to RE ($\alpha =  2$), making it the best suited for our perturbation-based corrections.
>
> We will integrate these clarifications into the revised manuscript.

---

### Meta-Review · Area_Chair_A3Tq · 2025-12-27

**Summary:**

The paper introduces a quantum-inspired framework for estimating uncertainty in Large Language Models (LLMs) to better detect hallucinations. Building on Semantic Entropy (Farquhar et al., 2024), the authors reinterpret the model’s sequence probabilities as a wave-function in a reproducing-kernel Hilbert space and construct a quantum tensor-network (QTN) representation to analyze perturbations in this space. They derive first-order perturbation features that quantify the local sensitivity of token probabilities and use these to re-weight output probabilities through an entropy-regularized optimization objective. In parallel, they compute semantic-cluster probabilities and a Renyi-entropy-based uncertainty score. Experiments across Mistral-7B, Falcon-1B, and LLaMA-2/3 on TriviaQA, NQ, SVAMP, and SQuAD show consistent AUROC improvements over Semantic Entropy, naive token entropy, and other baselines, including under 4-bit quantization.

**Reviewer Concerns:**

1. The proposed QTN embedding and perturbation analysis are computationally heavier than simpler uncertainty measures; inference-time cost is not reported.
2. The baselines should be broader
3. The technique details should be illustrated more clearly.

**Reviewer Scores:**

They should remain the scores

---

### Decision · Program_Chairs · 2026-01-26

Accept (Poster)